evolution/palaeontology

megaraptorids, *Australovenator*, Megaraptora, Winton Formation, dinosaurs, theropods

**Author for correspondence:**
Matt A. White
e-mail: fossilised@hotmail.com; mwhite62@une.edu.au

# New theropod remains and implications for megaraptorid diversity in the Winton Formation (lower Upper Cretaceous), Queensland, Australia

Matt A. White[1,2], Phil R. Bell[1], Stephen F. Poropat[2,3], Adele H. Pentland[2,3], Samantha L. Rigby[2,3], Alex G. Cook[2], Trish Sloan[2] and David A. Elliott[2]

[1]School of Environmental and Rural Science, University of New England, Armidale, New South Wales 2351, Australia
[2]Australian Age of Dinosaurs Natural History Museum, The Jump-Up, Winton, Queensland 4735, Australia
[3]Faculty of Science, Engineering and Technology, Swinburne University of Technology, John Street, Hawthorn, Victoria 3122, Australia

MAW, 0000-0002-4765-0356; PRB, 0000-0001-5890-8183

The holotype specimen of the megaraptorid *Australovenator wintonensis*, from the Upper Cretaceous Winton Formation (Rolling Downs Group, Eromanga Basin) of central Queensland, is the most complete non-avian theropod found in Australia to date. In fact, the holotype of *A. wintonensis* and isolated megaraptorid teeth (possibly referable to *Australovenator*) constitute the only theropod body fossils reported from the Winton Formation. Herein, we describe a new fragmentary megaraptorid specimen from the Winton Formation, found near the type locality of *A. wintonensis*. The new specimen comprises parts of two vertebrae, two metatarsals, a pedal phalanx and multiple unidentifiable bone fragments. Although the new megaraptorid specimen is poorly preserved, it includes the only megaraptorid vertebrae known from Queensland. The presence of pleurocoels and highly pneumatic caudal centra with camerate and camellate internal structures permit the assignment of these remains to Megaraptora gen. et sp. indet. A morphological comparison revealed that the distal end of metatarsal II and the partial pedal phalanx II-1 of the new specimen are morphologically divergent from *Australovenator*.

This might indicate the presence of a second megaraptorid taxon in the Winton Formation, or possibly intraspecific variation.

## 1. Introduction

Theropod discoveries in Australia are extremely rare and often constitute fragmentary and/or isolated bones. Consequently, their precise phylogenetic affinities have often proven difficult to determine with any certainty [1]. Although at least six Australian non-avian theropod taxa have been named, most of these are represented by only a single element and are regarded—although not always universally— as *nomina dubia*. These are: *Rapator ornitholestoides*, known only from a metacarpal I [2–4]; *Walgettosuchus woodwardi*, represented by a partial caudal vertebra [2,3]; *Kakuru kujani*, restricted to an incomplete tibia [3,5–7]; *Timimus hermani*, known only from a femur [3,8–12]; and *Ozraptor subotaii*, a distal tibia [3,6,13–20]. The only exception is *Australovenator wintonensis*, represented by a partial skeleton [4,21–26], which was initially classified as an indeterminate allosauroid but has since been universally allied with *Megaraptor* and its kin within Megaraptoridae [10,27,28]. Some general characteristics possessed by Megaraptoridae include: elongated three-digit hands with two enlarged recurved unguals on digits I and II and a much smaller digit III ungual [22,25,28], robust forearms [25,28], small blade-like teeth [9,23,29–33]; proportionally large feet compared to hind limb length and relatively gracile hindlimbs built for running [24,26]; and heavily pneumatized bones [29]. The completeness of the *Australovenator* type specimen has been fundamental to our current understanding of megaraptorid anatomy and phylogenetic hypotheses, and provided robust comparative data that have permitted the assignment of numerous isolated theropod specimens from the mid-Cretaceous of New South Wales [34,35] and Victoria [1,9] to Megaraptora (or its subclade Megaraptoridae), and validated an earlier report of a *Megaraptor*-like theropod from Victoria, based on an ulna [36]. The spatio-temporal range of Megaraptora (and Megaraptoridae) is becoming ever better understood as a result of numerous discoveries made within the last 2 decades. The South American record is the most extensive, diverse and abundant, with six taxa named to date: *Aoniraptor libertatem* [37], *Orkoraptor burkei* [33], *Megaraptor namunhuaiquii* [32,38,39], *Murusraptor barrosaensis* [30,40,41], *Aerosteon riocoloradensis* [28], and *Tratayenia rosalei* [42]. Numerous fragmentary specimens have also been reported from South America (see supplementary table 7 in [1]), which include the oldest (Albian [43]) and the youngest (Campanian [44]).

The Asian record of megaraptorans is steadily improving, with occurrences in Japan (*Fukuiraptor kitadaniensis*) [31,45], China (*Chilantaisaurus tashukouensis*) [46,47] and Thailand (*Phuwiangvenator yaemniyomi* and possibly *Vayuraptor nongbualamphuensis*) [48]. No megaraptorans are known from Antarctica, Europe or Africa (unless Bahariasauridae is a subclade of Megaraptora [37]) and only one taxon (*Siats meekerorum* [49]) is known from North America. Of these discoveries the best source of understanding megaraptoran skeletal anatomy have come from Australia and Argentina, thereby facilitating the identification of isolated and or fragmentary megaraptorid material. Herein, we describe the fragmentary remains of only the second megaraptorid specimen (excluding shed teeth) from the lower Upper Cretaceous Winton Formation near Winton, central Queensland, Australia. Morphological comparisons aided by three-dimensional software imaging were conducted in order to constrain its phylogenetic position, the implications of which are discussed herein.

## 2. Institutional abbreviations

Australian Age of Dinosaurs Museum of Natural History, Winton, Queensland, Australia (AAOD); Australian Age of Dinosaurs Fossil (AODF); Australian Age of Dinosaurs Locality (AODL); Museums Victoria (formerly National Museum of Victoria), Melbourne, Victoria, Australia (NMV).

## 3. Geological setting

The Winton Formation is the uppermost unit of the Eromanga Basin, a large continental basin that covers much of western Queensland [50]. The Winton Formation is transitional from the underlying marginal marine Mackunda Formation, with thin lenses of coastal and estuarine deposits persisting in the lower part of the formation [50] and dominated by sand- and mud-dominated facies representative of fluvial

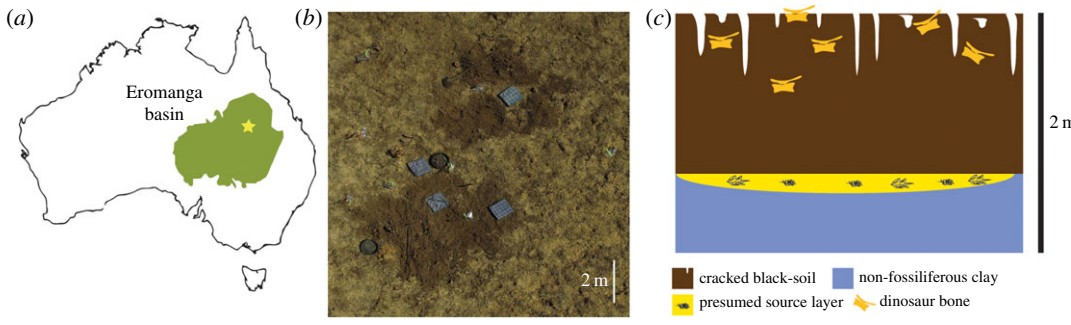

**Figure 1.** Locality and geological setting of AODL 261 (the 'Marilyn' Site). (*a*) Location of Elderslie Station (star) within the context of the Eromanga Basin (green), Central West Queensland, Australia. (*b*) Aerial photograph of AODL 261. (*c*) Schematic interpretation of the subsurface stratigraphy of AODL 261. Here, fossils are naturally brought to the surface from deeper fossiliferous horizons by the expansion–contraction of the clay-rich soils.

conditions in the upper (Cenomanian–lower Turonian) part of the formation [51,52]. The 'Marilyn' Site (AODL 261; nicknamed for its proximity to Mount Munro) was discovered and excavated on Elderslie Station in 2018, roughly 500 m west from the *Australovenator* type locality [21]. Rocks in this area are interpreted to come from the uppermost part of the Winton Formation close to the Cenomanian–Turonian boundary [52]. Surface exposures of the local geology are lacking in this area: the majority of specimens collected from this locality were exposed at or close to the surface within the montmorillonite-rich vertisol layer (colloquially termed 'black soil') that blankets the Winton Formation across much of the Winton Shire. Each bone fragment exposed on the surface was flagged prior to collection, so that the aereal extent of the specimens could be determined: the main concentration of bone occupied an area no more than 15 m². Vertebrate remains in this area are naturally exhumed from deeper (i.e. bedrock) layers by convective processes caused by the wetting/drying and the resulting swelling/contraction of the clay-rich soils. Deeper excavations at AODL 261 failed to recover additional remains; the layer presumed to be the source of the dinosaur remains was an approximately 5–10 cm thick layer of very fine sandy-clay with sporatic reworked plant fossils that were not formally identified. Below the plant-bearing layer was a barren, bluish-grey clay (greater than 1 m thick) entirely devoid of fossils (figure 1). Such clays, including those thought to have been the source of the current specimens, have been interpreted to represent low-energy fluvial deposits [4,21,23,53–56].

# 4. Material and methods

The *A. wintonensis* holotype specimens were computed tomography (CT) scanned at Queensland X-ray (Mackay Mater Hospital, Mackay, Queensland, Australia) using a Philips Brilliance CT 64-slice machine capable of producing 0.9 mm slices. Mimics v. 10.01 (Materialise HQ, Leuven, Belgium) was used to create the 3D surface meshes of the specimens. The meshes were exported as Binary *.stl files into Rhinoceros 5.0 (64-bit; Robert McNeal & Associates, Seattle, WA, USA), which was used to convert the files from *.stl to *.obj file format so they could be imported into Zbrush 4R7 P3 (Pixologic). The fragmentary megaraptorid specimens described herein were scanned using an Artec Space Spider 3D surface scanner.

The resulting 3D scans were exported as *.obj files so that they could be imported into Zbrush 4R7 P3 (Pixologic). Zbrush was used to digitally align and scale these specimens with the corresponding elements in *Australovenator* to confirm initial visual identification.

# 5. Systematic palaeontology

Theropoda Marsh, 1881 [57]
Tetanurae Gauthier, 1986 [58]
Coelurosauria von Huene 1914 [59]
Megaraptora Benson, Carrano & Brusatte, 2010 [27]
Megaraptoridae Novas, Agnolin, Ezcurra, Porfiri, Canale 2013 [10]
Megaraptoridae gen. et sp. indet.

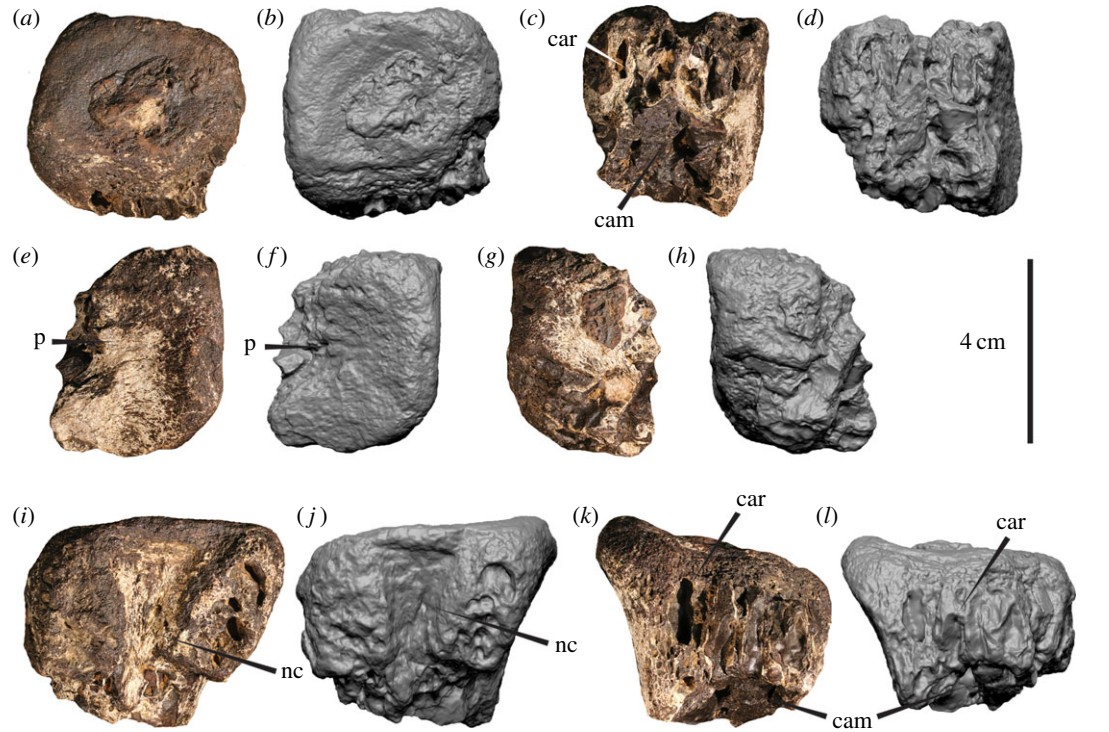

**Figure 2.** Megaraptorid caudal centrum (AODF 967) in (*a,b*) anterior, (*c,d*) posterior (*e,f*) right lateral, (*g,h*) left lateral (*i,j*) dorsal and (*k,l*) ventral views. car, camerate internat structure; cam, camellate internal structure; nc, neural canal; p, pleurocoel.

## 5.1. Material

Two incomplete caudal centra (AODF 967-968) (figures 2 and 3), proximal end of metatarsal II (AODF 977) (figure 4), distal end of metatarsal II (AODF 978) (figure 5), distal end of metatarsal IV (AODF 979) (figure 6), distal end of left pedal phalanx II-1 (AODF 972) (figure 7) and numerous unidentified fragments.

## 5.2. Locality

The 'Marilyn' Site (AODL 261), Elderslie Station, approximately 60 km NW of Winton, Queensland, Australia.

## 5.3. Horizon and Age

Uppermost Winton Formation, Rolling Downs Group, Eromanga Basin. Cenomanian–lowermost Turonian [51,52].

# 6. Results

## 6.1. Specimen descriptions

### 6.1.1. Vertebrae (AODF 967–968)

The likely positions of AODF 967 and AODF 968 within the vertebral series were estimated by comparisons with other megaraptorids [29,32,33,37,40,42]. Although incomplete, AODF 968 would likely have been longer than it is wide or tall (based in part on the presumed mid-centrum position of the pleurocoel; see below) with a nearly flat (anterior) endplate and no indication of paraphophyses. This combination of features is typical of caudal centra but unlike the anteroposteriorly short dorsal vertebrae and opisthocoeleous cervical vertebrae of megaraptorans [29,32,40]. The absence of chevron facets in AODF 968 further identifies it as the anterior part of the centrum. Overall, AODF 968

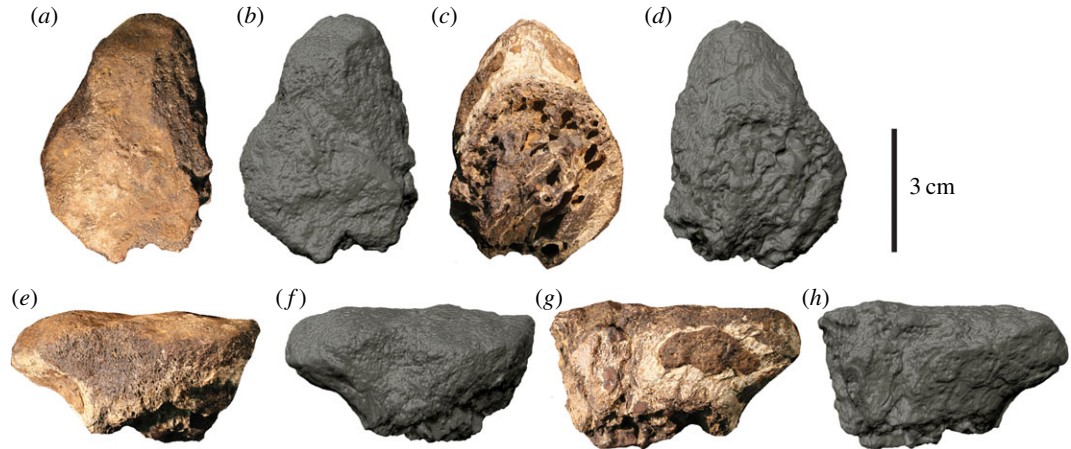

**Figure 3.** Megaraptorid caudal vertebra (AODF 968) in (a,b) posterior, (c,d) anterior, (e,f) right lateral, (g,h) left lateral, (i,j) dorsal and (k,l) ventral views. car, camerate internal structure; cam, camellate internal structure; nc, neural canal; p, pleurocoel.

**Figure 4.** Megaraptorid proximal left metatarsal II (AODF 977) in (a,b) proximal, (c,d) distal, (e,f) medial and (g,h) lateral views.

resembles the caudal vertebrae of the mid-caudal region of *Aerosteon riocoloradensis* (see fig. 9b in [29]). AODF 967 is less complete than AODF 968 and lacks the ventral edge of the centrum. Its proportions are therefore equivocal although its larger overall size suggests a more anterior position in the column than AODF 968 (table 1). The shallowly concave endplate and absence of parapophyses or sacral rib attachment scars eliminate a position in the cervical, anteriormost dorsal or sacral series.

Its relatively small size in comparison to the metapodials (table 1) suggest that it does not pertain to one of the dorsal vertebrae, which are typically much larger and have a stronger hour-glass shape than

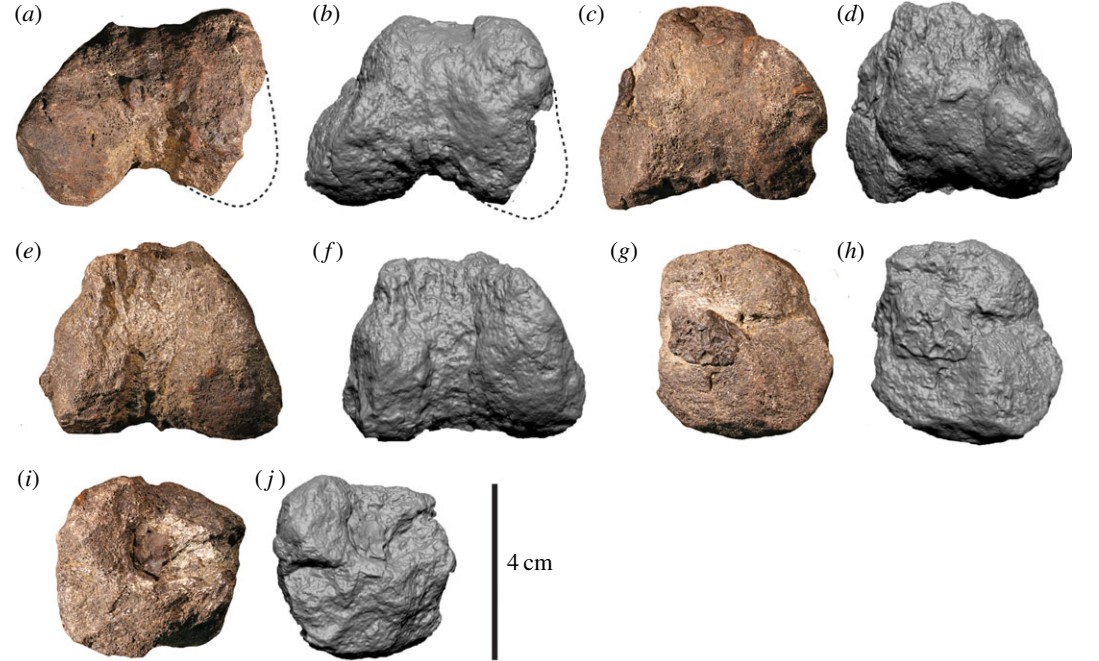

**Figure 5.** Megaraptorid distal right metatarsal IV (AODF 979) in (*a,b*) distal, (*c,d*) anterior, (*e,f*) posterior, (*g,h*) lateral and (*i,j*) medial views. Missing parts are reconstructed with a dashed line.

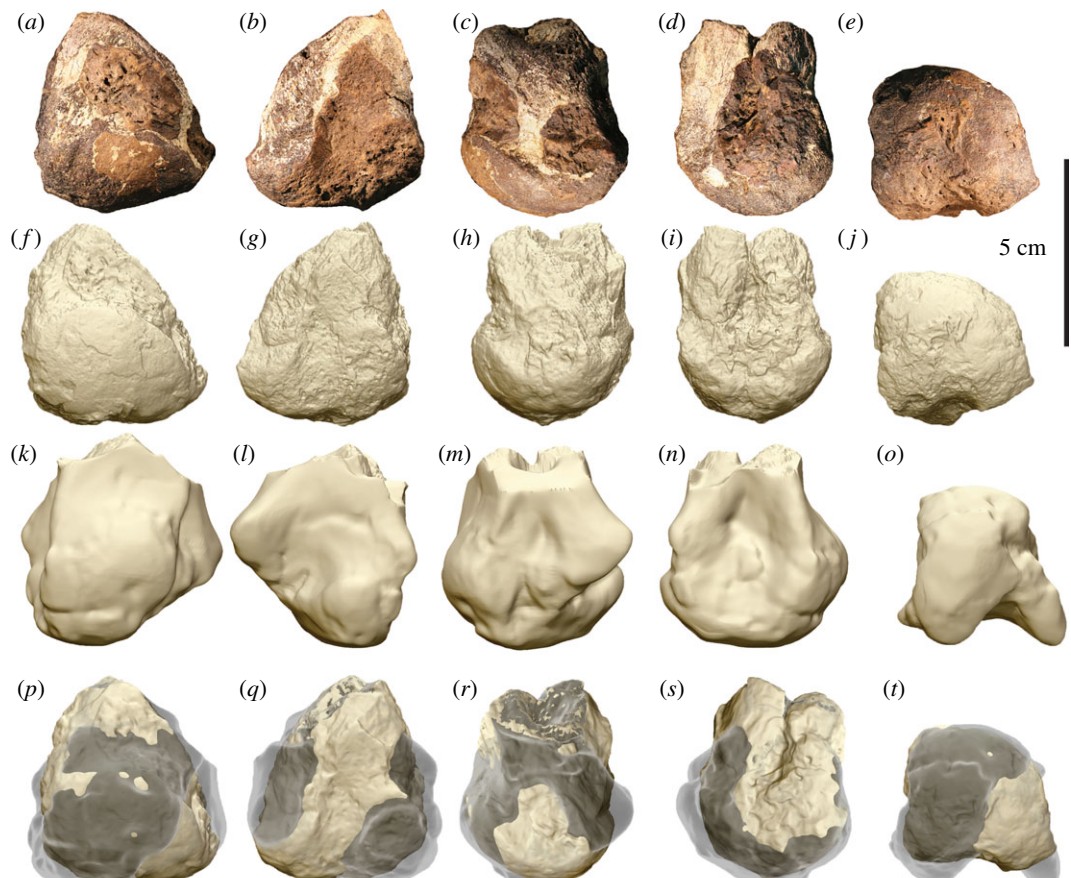

**Figure 6.** Megaraptorid distal right metatarsal II (AODF 978) compared with the right metatarsal II of *A. wintonensis* (AODF 604). Photographs (*a–e*) and digital renders (*f–j*) of megaraptorid right metatarsal II (AODF 978) in (*a,f*) anterior, (*b,g*) posterior, (*c,h*) medial, (*d,i*) lateral and (*e,j*) distal views. Digital renders (*k–o*) of *A. wintonensis* right metatarsal II (AODF 604) in (*k*) anterior, (*l*) posterior, (*m*) medial, (N) lateral and (*o*) distal views. Digital comparison (P–T) of right second metatarsals of AODF 978 (solid tan) with AODF 604 (*Australovenator*; transparent grey) corrected for scale and orientation in (*p*) anterior, (*q*) posterior, (*r*) medial, (*s*) lateral and (*t*) distal views.

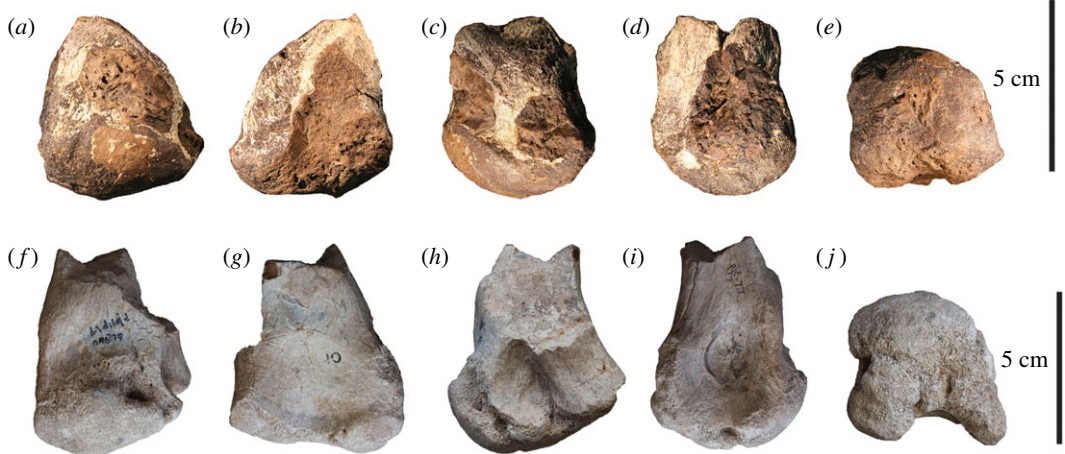

**Figure 7.** Megaraptorid distal right metatarsal II (AODF 978) compared with distal right metatarsal II of *Megaraptor* sp. (UNPSJB-PV 944). Photographs (*a–e*) of megaraptorid right metatarsal II (AODF 978) in (*a*) anterior, (*b*) posterior, (*c*) medial, (*d*) lateral and (*e*) distal views. Photographs (*f–j*) of *Megaraptor* sp. right metatarsal II (UNPSJB-PV 944) in (*f*) anterior, (*g*) posterior, (*h*) medial, (*i*) lateral and (*j*) distal views.

the caudal vertebrae (e.g. [32,42]). A caudal position more anterior than AODF 968 is therefore tenable. For descriptive purposes, AODF 967 is considered the anterior part of the centrum, although we concede that these attributions (i.e. the anterior part of a caudal centrum) are equivocal. AODF 967 constitutes the anterior portion of a vertebral centrum and lacking the ventral margin (figure 2). In posterior and ventral views, the broken surfaces reveal the camerate and camellate internal structures (figure 2*a,b,k,l*). The anterior articular surface (figure 2*c,d*) is shallowly concave and, when complete, would have been elliptical. Dorsally, the left and right neurocentral sutures are open (unfused), and their long axes extend anteromedially–posterolaterally (figure 2*i,j*). The mediolaterally concave neural canal is widest anteriorly, becoming narrower more posteriorly. Posterior to the articular endplate, the centrum is mediolaterally constricted, which, when complete, would depict an hour-glass shape in ventral view. The right lateral wall (which is more complete than the left side; figure 2*e,f*) preserves a small fossa approximately mid-height on the centrum, which appears to represent the posterior margin of a plurocoel. The ventral, right lateral and anterior surfaces are incomplete and poorly preserved, obscuring further morphological details of the centrum. AODF 968 comprises the anterior half of a caudal centrum (figure 3). The broken posterior surface reveals camerate and camellate internal structures (figure 3*a,b*). The anterior articular surface is elliptical (dorsoventrally taller than wide) and shallowly concave (figure 3*c,d*). The centrum is mediolaterally constricted posterior to the articular endplate, whereas the ventral edge (in lateral aspect) is nearly perpendicular to the endplate suggesting that the centrum was not notably dorsoventrally constricted. In right lateral view, there is a semicircular fossa situated close to the broken posterior edge at roughly two-thirds the height of the centrum and perforated by a pleurocoelous foramen (figure 3*e,f*). Directly posteromedial to this depression is a camerate internal structure resembling a pleurocoel. In dorsal view (figure 3*i,j*), the neural canal is mediolaterally concave and posteriorly tapering, bounded on either side by an anterolaterally–posteromedially oriented neurocentral suture.

The open sutural surfaces preserve numerous mediolaterally oriented grooves and ridges that would have reinforced the union with the pedicels of the corresponding neural arch. In ventral view (figure 3*k,l*), the centrum is transversely convex with no indication of a ventral groove or keel.

### 6.1.2. Proximal end of left metatarsal II (AODF 977)

The proximal end of a partial left metatarsal II is preserved. The proximal articular surface is somewhat pear shaped (narrowest posteriorly) and nearly flat (figure 4*a,b*). The proximal part of the metatarsal tapers immediately distal to the proximal articular surface, forming a shaft that is circular in cross-section (figure 4*c,d*), which is similar to the circular shaft in the metatarsal II of *Australovenator* (see fig. 7 in [24]). In medial (figure 4*e,f*) and lateral (figure 4*g,h*) views, the posterior margin is extended posteriorly relative to the preserved shaft. The medial margin is incomplete, exposing trabecular bone.

**Table 1.** Selected postcranial measurements of megaraptorid remains from AODL261 compared to *Australovenator*.

| specimens | AODF 967 (figure 2) | AODF 968 (figure 3) | AODF 977 (figure 4) | AODF 979 (figure 5) | AODF 978 (figure 6) Metatarsal II | *Australovenator* holotype AODF 604 (figure 6) Metatarsal II | AODF 972 (figure 7) MT II-1 | *Austrlovenator* holotype AODF 604 (figure 7) MT II-1 |
|---|---|---|---|---|---|---|---|---|
| centrum width at narrowest point | 30$^a$ | 23 | | | | | | |
| centrum height (to neurocentral suture) | 45$^a$ | 41 | | | | | | |
| centrum width (anterior end) | 46 | 36 | | | | | | |
| proximal width | | | 45$^a$ | | | | | |
| proximal height | | | 67$^a$ | | | | | |
| distal malleolus height (medial, lateral) | | | | 41$^a$, 42$^a$ | 44$^a$, 45$^a$ | 42, 40 | 30$^a$, 35$^a$ | 37, 33 |
| distal width (measured ventrally) | | | | 57$^a$ | 50$^a$ | 46 | — | 43 |

$^a$Specimens with broken or worn edges (representing incomplete measurements).

The lateral margin is more complete, laterally convex and distomedially inclined, providing an articular surface for metatarsal III.

### 6.1.3. Distal end of right metatarsal IV (AODF 979)

This specimen is interpreted as the distal end of a right metatarsal IV (AODF 979) based on comparisons with *Megaraptor* (see fig. 10 in [39]). In distal view (figure 5a,b), the lateral malleolus is inclined (approx. 70°) dorsomedially, whereas the medial malleolus is nearly vertical (approx. 5°). The borders of the medial and lateral collateral ligament pits are heavily eroded; nevertheless, the pits are distinguishable. The lateral malleolus is larger than the medial one and the two are seperated by a sulcus (flexor groove), which extends from the posterior (plantar) surface where it is deepest, to the anterior (dorsal) surface where it is comparatively shallow (figure 5a–d). This groove does not extend onto the short section of the preserved shaft nor is there any indication of an extensor pit proximal to the articular surface. The left metatarsal IV is present in the holotype of *Australovenator* (AODF 604), but its distal end is not preserved (see figs 9 and 10 in [24]), preventing any comparisons between the two.

### 6.1.4. Distal end of right metatarsal II (AODF 978)

Based on comparisons with *Australovenator* (see fig. 7 in [24]), AODF 978 is interpreted as the incomplete distal end of a right metatarsal II. The distal articular surface is nearly hemispherical but separated posteriorly (ventrally) into subequal medial and lateral malleoli by a broad, flexor groove. This groove is shallow, but likely misrepresented due to breakage and weathering of both medial and lateral malleolus. In anterior (dorsal) view (figure 6a,f), the distal condyle terminates proximally in a lip that borders a prominent extensor pit. In ventral (plantar) view, the medial malleolus extends further proximally than the lateral one. The medial malleolus is incomplete posteriorly and probably would have been somewhat longer still in life (based on comparisons with *Australovenator* (see fig. 7 in [24]). A shallow collateral ligament pit is present on the medial surface (figure 6c,h), whereas the lateral pit is deep but missing part of the ventral rim (figure 6d,i). Digitally superimposing AODF 978 with the distal end of metatarsal II of *Australovenator* helps to visualize a number of non-trivial differences (figure 6). The distal end of AODF 978 is more hemispherical in dorsal aspect than the strongly asymmetrical metatarsal II of *Australovenator* (figure 6p,q). More specifically, the medial malleolus of *Australovenator* is proximally positioned relative to the medial malleolus, mediolaterally compressed and bladelike (figure 6t). By contrast, the medial malleolus of AODF 978 falls along the same transverse plane as the lateral malleolus (in posterior aspect; figure 6g) and, despite being incomplete, is relatively robust. In posterior view, the sulcus separating the malleoli is shallower in AODF 978 than in *Australovenator* (figure 6q), although this may be exaggerated by breakage/weathering in the former. Additionally, AODF 978 is distinctly larger than *Australovenator* (table 1). Intriguingly, the distal end of metatarsal II (AODF 978) closely resembles the same element (UNPSJB-Pv944) that was tentatively assigned to *Megaraptor* sp. [60] (figure 7) from the roughly coeval Bajo Barreal Formation (Chubut Group, Golfo de San Jorge Basin) of Chubut Province, Argentina, rather than *Australovenator*.

In particular, both specimens share a distal articular surface that is somewhat hemispherical with medial and lateral malleoli that fall along the same transverse plane (or nearly so in the case of UNPSJB-Pv944) in ventral aspect. In distal view, the flexor groove separating the medial and lateral malleoli is relatively shallow (although possibly an artefact, accentuated in AODF 978 by breakage) compared to *Australovenator*. The weathering suffered by AODF 978 precludes any useful comparisons of the medial or lateral surfaces. Unfortunately, a transparent overlay could not be replicated for the UNPSJB-PV 944 specimen as a 3D surface mesh has not yet been developed for the specimen.

### 6.1.5. Distal end of left pedal phalanx II-1 (AODF 972)

The sole pedal phalanx (AODF 972) recovered from AODL 261 is interpreted as left II-1, based on comparisons with *Australovenator* [24,26]; however, due to the specimen's incompleteness this identification is tentative. The specimen consists of the distal articular end and a short section of the shaft, which is subcircular in cross-section and hollow. The distal articular surface is ginglymous, dorsoventrally and, to a lesser extent, mediolaterally expanded relative to the shaft (figure 8e,j). Although broken, the medial condyle is dorsoventrally shorter than the lateral one but roughly equal in mediolateral width (table 1). The collateral ligament pits, while present, are infilled with ironstone (figure 8c,d,h,i). This element does not differ notably from that of *Australovenator* (figure 8p–t). Minor

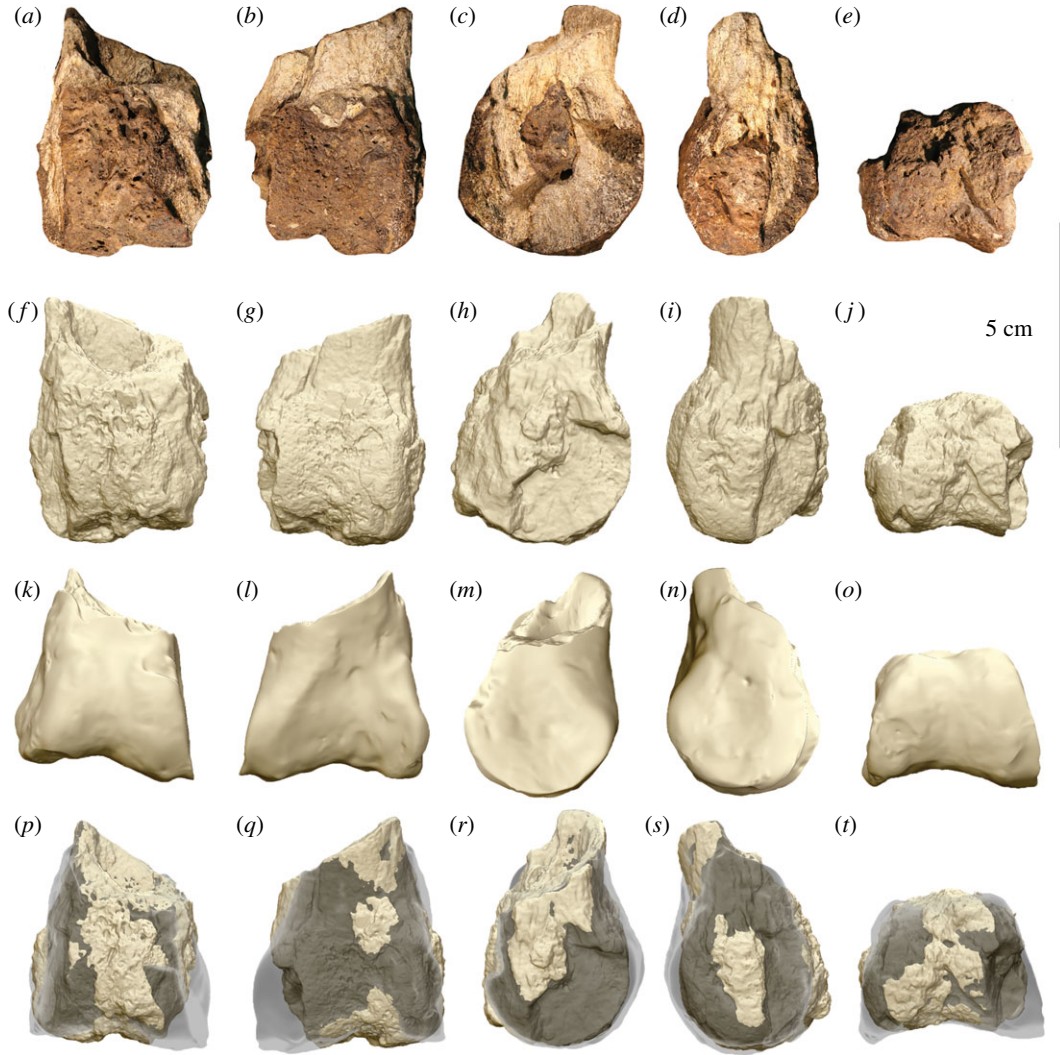

**Figure 8.** Megaraptorid distal left pedal phalanx II-1 (AODF 972) compared with the left pedal phalanx II-1 of *A. wintonensis* (AODF 604). Photographs (*a*–*e*) and digital renders (*f*–*j*) of megaraptorid left pedal phalanx II-1 (AODF 972) in (*a,f*) anterior, (*b,g*) posterior, (*c,h*) medial, (*d,i*) lateral and (*e,j*) distal views. Digital renders (*k*–*o*) of *A. wintonensis* left pedal phalanx II-1 (AODF 604) in (*k*) anterior, (*l*) posterior, (*m*) medial, (*n*) lateral and (*o*) distal views. Digital comparison (*p*–*t*) of left pedal phalanges II-1 of AODF 978 (solid tan) with AODF 604 (*Australovenator*; transparent grey) in (*p*) anterior, (*q*) posterior, (*r*) medial, (*s*) lateral and (*t*) distal views.

areas of morphological discrepancy can be attributed to breakage and/or the adherent ironstone matrix (figure 8*a*–*e*).

## 7. Discussion

Fragmentary theropod remains recovered from the 'Marilyn' Site (AODL 261) constitute only the second theropod specimen (excluding shed teeth) from the Winton Formation. The close proximity and size congruence of the specimens recovered from AODL 261 suggests that they pertain to a single individual.

Unfortunately, our failure to locate further theropod remains in the suspected source layer precludes identification of the taphonomic processes to which these bones were subjected prior to their exposure: the effects of all such processes have been overprinted by much more recent weathering. Identifiable elements are limited in number—two fragmentary vertebrae, three partial metatarsals and the distal end of a pedal phalanx—and all were significantly weathered. The poor preservation and lack of diagnostic features among the described specimens prevented a meaningful phylogenetic analysis from being undertaken. Nevertheless, some characters typical of megaraptorids—including camerate and camellate vertebral centra [32,35] and the presence of pleurocoels [30]—are both evident in AODF

967 and AODF 968. The identification of the AODL 261 material as megaraptorid lies principally on the presence of pleurocoels on the two incomplete caudal centra. Pleurocoels are uncommon in the caudal vertebrae of theropods [33]. Although few megaraptorid caudal vertebrae are known, pleurocoels are present in *Aerosteon* [29], *Megaraptor* [32], *Orkoraptor* [33] and *Aoniraptor* [37]. Caudal pleurocoels are absent in the immediate outgroups to Megaraptoridae (e.g. *Fukuiraptor* [45], unknown in *Chilantaisaurus*) as well as *Neovenator* [61,62] but are present in the megalosaurid *Torvosaurus* [63], the carcharodontosaurid *Carcharodontosaurus* [64] and oviraptorosaurs, none of which have been unambiguously identified from Australia [3,9]. The distal end of metatarsal II (AODF 978) also bears some resemblance to that of a specimen assigned to *Megaraptor* sp. (UNPSJB-PV 944 [60]) and to a lesser extent *Australovenator* [24]. It alone is not diagnostic enough to identify as a megaraptorid; however, alongside the pleurocoelus caudal vertebrae, its dimensions suggest the individual to which they pertained was slightly larger than the *A. wintonensis* type individual (AODF 604), and that it was possibly similar in size to the largest megaraptorids known from Victoria (NMV P186153) [1] and New South Wales [34]. Our preliminary results indicate that these remains belong to Megaraptoridae indet., from the Winton Formation based primarily on the distal end of metatarsal II; however, more complete and better-preserved material is required to establish this claim.

## 8. Conclusion

This paper describes the fragmentry remains of only the second non-avian theropod skeleton recovered from the Winton Formation in Central Queensland, Australia. The remains, presumed to have come from a single individual, are assigned to Megaraptoridae indet. based on the presence of camerate and camellate internal structures and the presence of pleurocoels in caudal vertebrae. Given the size of the distal ends of metatarsal II (AODF 972) and IV (AODF 979), this individual would have been larger than the holotype of *Australovenator* (AODF 604). Additionally, morphological discrepancies between *Australovenator* and the new specimens maybe representative of either ontogenetic/intraspecific variation or indicative of the presence of a second megaraptorid from the Winton Formation. If correct, the latter interpretation adds further support to previous claims [1,34] that megaraptorids were the dominant large predator in many Australian mid-Cretaceous terrestrial ecosystems.

Data accessibility. The data that support the findings of this study are available from the corresponding author M.A.W. with the permission of Australian Age of Dinosaurs Museum. Restrictions apply to the availability of these data, which were used under licence for this study.

Authors' contributions. M.A.W. and P.R.B. designed the project. M.A.W., P.R.B and S.F.P. described the specimens. M.A.W. collected synchrotron data and assembled the figures. M.A.W., P.R.B., S.F.P., A.H.P., S.L.R. and A.G.C. contributed to writing of the paper. M.A.W., T.S. and D.A.E. oversaw the collection, preparation and curation of the fossils. M.A.W., P.R.B., S.F.P., A.H.P., S.L.R., A.G.C., T.S. and D.A.E. approve the version to be published. M.A.W., P.R.B., S.F.P., A.H.P., S.L.R., A.G.C., T.S. and D.A.E. agree to be accountable for all aspects of the work in ensuring that questions related to the accuracy or integrity of any part of the work are appropriately investigated and resolved.

Competing interests. We declare we have no competing interests.

Funding. Funding for this work was provided to M.A.W. by the UNE Post-Doctoral Research Fellow Operating Grant provided by the University of New England, Armidale; ANSTO for Imaging and Medical beamline usage award (project ID: AS183/IMBL/13963) to M.A.W.; Australian Research Council Discovery Early Career Researcher Award (project ID: DE170101325) to P.R.B.

Acknowledgements. The authors thank the Australian Age of Dinosaurs Museum, Queensland, Australia, and Museums Victoria, Melbourne, Australia, for access to the specimens. The authors are grateful to all of the Australian Age of Dinosaur participants who attended the 2018 dig season and helped with the excavation. Thanks to the reviewers Alexis Mauro Aranciaga Rolando and the anonymous reviewer for their helpful comments. Thanks to Dr Joseph Bevitt and Dr Anton Maksimenko for assistance in data collection at ANSTO. Thanks to Rubén Martínez for giving permission to image Megaraptor sp. (UNPSJB-PV 944) as part of our research. Thanks to the site manager Robert A. Elliott for managing the excavation of dinosaur material and driving the excavation equipment.

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
