## [Reviewer comments · Royal Society Open Science]

Review History

RSOS-191462.R0 (Original submission)

Review form: Reviewer 1 (Fernando Novas)

Is the manuscript scientifically sound in its present form?

Yes

Are the interpretations and conclusions justified by the results?

No

Is the language acceptable?

Yes

Do you have any ethical concerns with this paper?

No

Have you any concerns about statistical analyses in this paper?

No

Recommendation?

Accept with minor revision (please list in comments)

Comments to the Author(s)

REVIEW of "New theropod remains and implications for megaraptorid diversity in the Winton Formation (lower Upper Cretaceous), Queensland, Australia", by Matt White and co-authors.

Let me suggest the following corrections:

ABSTRACT

Line 50. Title and abstract refer to "megaraptorid" (derived from family name Megaraptoridae). Moreover, authors offer anatomical basis to refer the new specimen to this family. Then, it is required to be consistent with this, replacing "Megaraptora" for "Megaraptoridae".

Lines 51-58. I highly recommend to change this last paragraph for a more cautious one. See comments below on this same aspect.

INTRODUCTION

PAGE 3 - Line 28. Reference 10 (Novas et al., 2013) must be added here, because it includes a response to Benson et al. 2009.

PAGE 3 - Line 57. Datations obtained by Sickmann et al. strongly support a Campanian age for Cerro Fortaleza Formation, and then for Orkoraptor.

PAGE 3 - Line 58. I recommend remove this comment, because Orkoraptor was originally thought as Maastrichtian, which is now considered wrong.

PAGE 4 - Authors say that "These discoveries have provided a much-improved understanding of megaraptoran skeletal anatomy, thereby facilitating the identification of isolated megaraptorid material". No doubt Australian (i.e., Australovenator), Patagonian (different megaraptorid species) and Japanese (Fukuiraptor) specimens conform the best (and probably the only one) source of information about megaraptoran anatomy and evolution. However, the remaining taxa cited here as "Asian and North American megaraptorans", are not members of this clade. As extensively discussed in previous papers (e.g., Novas et al., 2013, 2016; Porfiri et al., 2014; Aranciaga Rolando et al., 2019) Chilantaisaurus does not exhibit synapomorphies of Megaraptoridae. The same comments apply for the poorly known Siats, the morphology of which is sharply different from that of megaraptorans. Regarding Thai coelurosaurians Phuwianvenator and Vayuraptor, they show some similarities with megaraptorans, although they can be also interpreted because these two Thai theropods and megaraptorans are basal coelurosaurians. In sum, let me repeat: best source for megaraptorid anatomy comes from Australia and Argentina.

SYSTEMATIC PALEONTOLOGY

PAGE 6 -TETANURAE - Increasing morphological evidence bolsters Megaraptora as basal Coelurosauria (see Novas et al., 2013, 2016; Porfiri et al., 2014, 2018; Aranciaga Rolando et al., 2019; Apesteguía et al., 2016). By citing Tetanurae alone, this Systematic Paleontology section oversees such information.

PAGE 6 - MEGARAPTORA - The term "Megaraptoridae/megaraptorid" is used in the title, abstract, and main text of the present ms. Moreover, authors conclude (on the basis of synapomorphies) that the new specimen is a member of Megaraptoridae. Then, authors must insert this family name as part of Systematic Paleontology.

PAGE 7 - Lines 34-36. It is appropriate the use of any well-known theropod (let say: Allosaurus, Tyrannosaurus, Deinonychus, Neovenator) to identify isolated bone materials. However, the old (2009) idea that the allosauroid Neovenator is closely related with coelurosaurian megaraptorans has been demonstrated wrong in several recent contributions (see Novas et al. 2013; Porfiri et al., 2014; Aranciaga Rolando et al., 2019).

PAGE 7 - Lines 45 and 52. Authors are sure of the megaraptoran affiliation of the available bones. Then, it seems enough compare with "other megaraptorans". Citing of Neovenator here is not required (even theropods other than megaraptorans).

PAGE 9 - Lines 31 and 32. The bone illustrated on figure 5 of present ms, seems fairly symmetrical in distal view, being similar to mtt III as well as to some non-ungual pedal phalanges. On the contrary, metatarsal IV in theropods is beveled in distal view. Please, have a look at Calvo et al., 2004 paper (your reference number 35), figure 10 A,C, illustrating the mtt IV of Megaraptor namunhuaiquii. Also, consider the possibility that this bone corresponds to a pedal phalanx.

PAGE 10. Lines 51-54. Are you meaning that the specimen here described resembles more Megaraptor sp from Patagonia, rather than to Australovenator? Please, clarify. However, and based on the figures here afforded, distal end of mtt II of Australovenator matches well with that of Megaraptor sp. from Patagonia, both being different from the new Megaraptoridae from Winton in the ventral projection of both inner and outer condyles. Then, let me ask whether shape "distinctions" of the later one are due to erosion, rather than true anatomical features.

PAGE 10. Lines 56. Please, replace "Golfo do" for "Golfo de".

PAGE 12. Lines 55-60. Authors offer a "preliminary result" recognizing a second megaraptorid taxon in Winton Fm. However, such conclusion is based on highly incomplete and fragmentary evidence. I recommend to refer the new specimen as Megaraptoridae indet., thus avoiding inflation of taxonomic diversity on the basis of tenuous evidence.

PAGE 13. Lines 1-2. Two comments about this statement: first, although other sites may show sympatric megaraptorans, this does not automatically support the presence of sympatric megaraptorans in Winton. Second, the case from Thailand is the same: Vayuraptor is based on minor anatomical distinctions with respect to Phuwiangvenator. As a final comment, morphological variation within population of a single species is an almost ignored topic in theropod papers.

PAGE 13. Lines 15. Is this taxonomically correct? Or it may be "Megaraptoridae indet."?

PAGE 13. Lines 25-26. Please, remove this phrase. It is not the software what allows recognizing distinctions.

PAGE 13. Lines 27. I completely agree with this cautious and reasonable expression. Authors must apply this view in the rest of the text.

PAGE 13. Lines 29-34. I completely agree with authors in this regard, even recognizing a single megaraptorid species for the Winton Formation.

PAGE 13. Lines 31. Please, detail reference source for such "previous claims"

Review form: Reviewer 2 (Mauro Aranciaga Rolando)

Is the manuscript scientifically sound in its present form?

Yes

Are the interpretations and conclusions justified by the results?

Yes

Is the language acceptable?

Yes

Do you have any ethical concerns with this paper?

No

Have you any concerns about statistical analyses in this paper?

No

Recommendation?

Accept with minor revision (please list in comments)

Comments to the Author(s)

Dear authors and editor of the Royal Society Open Science:

The contribution is well-written, interesting and useful. I have attached a PDF file (Appendix A) with some minor comments. I am very happy to say that this paper is able to be published in the Royal Society Open Science.

Yours sincerely,

Alexis Mauro Aranciaga Rolando

Decision letter (RSOS-191462.R0)

22-Nov-2019

Dear Dr White,

On behalf of the Editors, I am pleased to inform you that your Manuscript RSOS-191462 entitled "New theropod remains and implications for megaraptorid diversity in the Winton Formation (lower Upper Cretaceous), Queensland, Australia" has been accepted for publication in Royal Society Open Science subject to minor revision in accordance with the referee suggestions. Please find the referees' comments at the end of this email.

The reviewers and handling editors have recommended publication, but also suggest some minor revisions to your manuscript. Therefore, I invite you to respond to the comments and revise your manuscript. The suggested minor corrections are important and will improve your manuscript, but will not take you long to perform. If there are any changes that you do not wish to make, please explain clearly your reasons.

- Ethics statement

- Data accessibility

<http://datadryad.org/submit?journalID=RSOS&manu=RSOS-191462>

- Competing interests

- Authors' contributions

- Acknowledgements

- Funding statement

Please ensure you have prepared your revision in accordance with the guidance at <https://royalsociety.org/journals/authors/author-guidelines/> -- please note that we cannot publish your manuscript without the end statements. We have included a screenshot example of

the end statements for reference. If you feel that a given heading is not relevant to your paper, please nevertheless include the heading and explicitly state that it is not relevant to your work.

Because the schedule for publication is very tight, it is a condition of publication that you submit the revised version of your manuscript before 01-Dec-2019. Please note that the revision deadline will expire at 00.00am on this date. If you do not think you will be able to meet this date please let me know immediately.

Please note that Royal Society Open Science charge article processing charges for all new submissions that are accepted for publication. Charges will also apply to papers transferred to Royal Society Open Science from other Royal Society Publishing journals, as well as papers

submitted as part of our collaboration with the Royal Society of Chemistry (<https://royalsocietypublishing.org/rsos/chemistry>).

If your manuscript is newly submitted and subsequently accepted for publication, you will be asked to pay the article processing charge, unless you request a waiver and this is approved by Royal Society Publishing. You can find out more about the charges at <https://royalsocietypublishing.org/rsos/charges>. Should you have any queries, please contact openscience@royalsociety.org.

Kind regards,

on behalf of Dr Julia Brenda Desojo (Associate Editor) and Jon Blundy (Subject Editor)
openscience@royalsociety.org

Reviewer comments to Author:

Reviewer: 1
Comments to the Author(s)

REVIEW of "New theropod remains and implications for megaraptorid diversity in the Winton Formation (lower Upper Cretaceous), Queensland, Australia", by Matt White and co-authors.

Let me suggest the following corrections:

ABSTRACT

Line 50. Title and abstract refer to "megaraptorid" (derived from family name Megaraptoridae). Moreover, authors offer anatomical basis to refer the new specimen to this family. Then, it is required to be consistent with this, replacing "Megaraptora" for "Megaraptoridae".

Lines 51-58. I highly recommend to change this last paragraph for a more cautious one. See comments below on this same aspect.

INTRODUCTION

PAGE 3 - Line 28. Reference 10 (Novas et al., 2013) must be added here, because it includes a response to Benson et al. 2009.

PAGE 3 - Line 57. Datations obtained by Sickmann et al. strongly support a Campanian age for Cerro Fortaleza Formation, and then for Orkoraptor.

PAGE 3 - Line 58. I recommend remove this comment, because Orkoraptor was originally thought as Maastrichtian, which is now considered wrong.

PAGE 4 - Authors say that "These discoveries have provided a much-improved understanding of megaraptoran skeletal anatomy, thereby facilitating the identification of isolated megaraptorid material". No doubt Australian (i.e., Australovenator), Patagonian (different megaraptorid

species) and Japanese (Fukuiraptor) specimens conform the best (and probably the only one) source of information about megaraptoran anatomy and evolution. However, the remaining taxa cited here as “Asian and North American megaraptorans”, are not members of this clade. As extensively discussed in previous papers (e.g., Novas et al., 2013, 2016; Porfiri et al., 2014; Aranciaga Rolando et al., 2019) *Chilantaisaurus* does not exhibit synapomorphies of Megaraptoridae. The same comments apply for the poorly known *Siats*, the morphology of which is sharply different from that of megaraptorans. Regarding Thai coelurosaurians *Phuwianvenator* and *Vayuraptor*, they show some similarities with megaraptorans, although they can be also interpreted because these two Thai theropods and megaraptorans are basal coelurosaurs. In sum, let me repeat: best source for megaraptorid anatomy comes from Australia and Argentina.

SYSTEMATIC PALEONTOLOGY

PAGE 6 -TETANURAE - Increasing morphological evidence bolsters Megaraptora as basal Coelurosauria (see Novas et al., 2013, 2016; Porfiri et al., 2014, 2018; Aranciaga Rolando et al., 2019; Apesteguía et al., 2016). By citing Tetanurae alone, this Systematic Paleontology section oversees such information.

PAGE 6 - MEGARAPTORA - The term "Megaraptoridae/megaraptorid" is used in the title, abstract, and main text of the present ms. Moreover, authors conclude (on the basis of synapomorphies) that the new specimen is a member of Megaraptoridae. Then, authors must insert this family name as part of Systematic Paleontology.

PAGE 7 - Lines 34-36. It is appropriate the use of any well-known theropod (let say: *Allosaurus*, *Tyrannosaurus*, *Deinonychus*, *Neovenator*) to identify isolated bone materials. However, the old (2009) idea that the allosauroid *Neovenator* is closely related with coelurosaurian megaraptorans has been demonstrated wrong in several recent contributions (see Novas et al. 2013; Porfiri et al., 2014; Aranciaga Rolando et al., 2019).

PAGE 7 - Lines 45 and 52. Authors are sure of the megaraptoran affiliation of the available bones. Then, it seems enough compare with "other megaraptorans". Citing of *Neovenator* here is not required (even theropods other than megaraptorans).

PAGE 9 - Lines 31 and 32. The bone illustrated on figure 5 of present ms, seems fairly symmetrical in distal view, being similar to mtt III as well as to some non-ungual pedal phalanges. On the contrary, metatarsal IV in theropods is beveled in distal view. Please, have a look at Calvo et al., 2004 paper (your reference number 35), figure 10 A,C, illustrating the mtt IV of *Megaraptor namunhuaiquii*. Also, consider the possibility that this bone corresponds to a pedal phalanx.

PAGE 10. Lines 51-54. Are you meaning that the specimen here described resembles more *Megaraptor* sp from Patagonia, rather than to *Australovenator*? Please, clarify. However, and based on the figures here afforded, distal end of mtt II of *Australovenator* matches well with that of *Megaraptor* sp. from Patagonia, both being different from the new Megaraptoridae from Winton in the ventral projection of both inner and outer condyles. Then, let me ask whether shape “distinctions” of the later one are due to erosion, rather than true anatomical features.

PAGE 10. Lines 56. Please, replace “Golfo do” for "Golfo de ".

PAGE 12. Lines 55-60. Authors offer a "preliminary result" recognizing a second megaraptorid taxon in Winton Fm. However, such conclusion is based on highly incomplete and fragmentary evidence. I recommend to refer the new specimen as Megaraptoridae indet., thus avoiding inflation of taxonomic diversity on the basis of tenuous evidence.

PAGE 13. Lines 1-2. Two comments about this statement: first, although other sites may show sympatric megaraptorans, this does not automatically support the presence of sympatric megaraptorans in Winton. Second, the case from Thailand is the same: Vayuraptor is based on minor anatomical distinctions with respect to Phuwiangvenator. As a final comment, morphological variation within population of a single species is an almost ignored topic in theropod papers.

PAGE 13. Lines 15. Is this taxonomically correct? Or it may be "Megaraptoridae indet."?

PAGE 13. Lines 25-26. Please, remove this phrase. It is not the software what allows recognizing distinctions.

PAGE 13. Lines 27. I completely agree with this cautious and reasonable expression. Authors must apply this view in the rest of the text.

PAGE 13. Lines 29-34. I completely agree with authors in this regard, even recognizing a single megaraptorid species for the Winton Formation.

PAGE 13. Lines 31. Please, detail reference source for such "previous claims"

Reviewer: 2

Comments to the Author(s)

Dear authors and editor of the Royal Society Open Science:

The contribution is well-written, interesting and useful. I have attached a PDF file with some minor comments. I am very happy to say that this paper is able to be published in the Royal Society Open Science.

Yours sincerely,

Alexis Mauro Aranciaga Rolando

Author's Response to Decision Letter for (RSOS-191462.R0)

See Appendix B.

Decision letter (RSOS-191462.R1)

28-Nov-2019

Dear Dr White,

It is a pleasure to accept your manuscript entitled "New theropod remains and implications for

megaraptorid diversity in the Winton Formation (lower Upper Cretaceous), Queensland, Australia" in its current form for publication in Royal Society Open Science. The comments of the reviewer(s) who reviewed your manuscript are included at the foot of this letter.

on behalf of Dr Julia Brenda Desojo (Associate Editor) and Jon Blundy (Subject Editor)
openscience@royalsociety.org

Appendix A**ROYAL SOCIETY
OPEN SCIENCE****New theropod remains and implications for megaraptorid
diversity in the Winton Formation (lower Upper
Cretaceous), Queensland, Australia**

Journal:	Royal Society Open Science
Manuscript ID	RSOS-191462
Article Type:	Research
Date Submitted by the Author:	30-Aug-2019
Complete List of Authors:	White, Matt; University of New England, Environmental and Rural Science; Australian Age of Dinosaurs Museum of Natural History, Palaeontology Bell, Phil; University of New England, Environmental and Rural Science; 4 Pipestone Creek Dinosaur Initiative Poropat, Stephen; Swinburne University of Technology Faculty of Science Engineering and Technology; Australian Age of Dinosaurs Natural History Museum, Cook, Alex; Australian Age of Dinosaur Museum of Natural History Pentland, Adele; Swinburne University of Technology Faculty of Science Engineering and Technology; Australian Age of Dinosaurs Museum of Natural History, Palaeontology Rigby, Samantha; Swinburne University of Technology Faculty of Science Engineering and Technology; Australian Age of Dinosaurs Museum of Natural History Sloan, Trish; Australian Age of Dinosaurs Museum of Natural History, Palaeontology Elliott, David; Australian Age of Dinosaurs Museum of Natural History, Palaeontology
Subject:	evolution < BIOLOGY, palaeontology < BIOLOGY
Keywords:	megaraptorids, Australovenator wintonensis, Winton Formation, theropod, dinosaur
Subject Category:	Earth science

Author-supplied statements

Relevant information will appear here if provided.

Ethics

Does your article include research that required ethical approval or permits?:

This article does not present research with ethical considerations

Statement (if applicable):

CUST_IF_YES_ETHICS :No data available.

Data

It is a condition of publication that data, code and materials supporting your paper are made publicly available. Does your paper present new data?:

Yes

Statement (if applicable):

The 3D data that support the findings of this study are available from the Dryad Digital Repository:

<https://doi.org/10.5061/dryad.4q5j211>

Reviewer URL: <https://datadryad.org/review?doi=doi:10.5061/dryad.4q5j211>

Conflict of interest

I/We declare we have no competing interests

Statement (if applicable):

CUST_STATE_CONFLICT :No data available.

Authors' contributions

This paper has multiple authors and our individual contributions were as below

Statement (if applicable):

MAW, PRB designed the project. MAW, PRB, SFP described the specimens. MAW assembled the figures. MAW, PRB, SFP, AHP, SLR, AGC contributed to writing of the paper. MAW, TS, DAE oversaw the collection, preparation and curation of the fossils. MAW, PRB, SFP, AHP, SLR, AGC, TS, DAE approve of the version to be published. MAW, PRB, SFP, AHP, SLR, AGC, TS, DAE agree to be accountable for all aspects of the work in ensuring that questions related to the accuracy or integrity of any part of the work are appropriately investigated and resolved.

New theropod remains and implications for megaraptorid diversity in the Winton Formation (lower Upper Cretaceous), Queensland, Australia

Matt A. White^{1,2}; Phil R. Bell¹; Stephen F. Poropat^{2,3}; Adele H. Pentland^{2,3}; Samantha L. Rigby^{2,3}; Alex G. Cook²; Trish Sloan²; David A. Elliott²

¹School of Environmental & Rural Science, University of New England, Armidale, New South Wales 2351, Australia

²Australian Age of Dinosaurs Natural History Museum, The Jump-Up, Winton, Queensland 4735, Australia

³Faculty of Science, Engineering and Technology, Swinburne University of Technology, John St, Hawthorn, Victoria 3122, Australia

Keywords: Megaraptorids, *Australovenator*, Megaraptora, Winton Formation, dinosaurs, theropods

1. Summary

The holotype specimen of the megaraptorid *Australovenator wintonensis*, from the Upper Cretaceous Winton Formation (Rolling Downs Group, Eromanga Basin) of central Queensland, is the most complete non-avian theropod found in Australia to date. In fact, the holotype of *A. wintonensis* and isolated megaraptorid teeth (possibly referable to *Australovenator*) constitute the only theropod body fossils reported from the Winton Formation. Herein, we describe a new fragmentary megaraptorid specimen from the Winton Formation, found near the type locality of *A. wintonensis*. The new specimen comprises parts of two vertebrae, two metatarsals, a pedal phalanx and multiple unidentifiable bone fragments. Although the new megaraptorid specimen is poorly preserved, it includes the only megaraptorid vertebrae known from Queensland. The presence of pleurocoels and highly pneumatic caudal centra with camerate and camellate internal structures, permit the assignment of these remains to Megaraptora gen. et sp. indet. A morphological comparison revealed that the distal end of metatarsal II and the partial pedal phalanx II-1 of the new specimen are morphologically divergent from *Australovenator*. This might indicate the presence of a second megaraptorid taxon in the Winton Formation, or possibly intraspecific variation.

*Author for correspondence (Matt A White: fossilised@hotmail.com; mwhite62@une.edu.au)

†Present address: Environmental and Rural Science, University of New England, Armidale NSW, 2351, Australia

2. Introduction

Theropod discoveries in Australia are extremely rare and often constitute fragmentary and/or isolated bones. Consequently, their precise phylogenetic affinities have often proven difficult to determine with any certainty [1]. Although at least six Australian non-avian theropod taxa have been named, most of these are represented by only a single element and are regarded—although not always universally—as *nomina dubia*. These are: *Rapator ornitholestoides*, known only from a metacarpal I [2-4]; *Walgettosuchus woodwardi*, represented by a partial caudal vertebra [2, 3]; *Kakuru kujani*, restricted to an incomplete tibia [3, 5-7]; *Timimus hermani*, known only from a femur [3, 8-12]; and *Ozraptor subotaii*, a distal tibia [3, 6, 13-20]. The only exception is *Australovenator wintonensis*, represented by a partial skeleton [4, 21-26], which was initially classified as an indeterminate allosauroid but has since been universally allied with *Megaraptor* and its kin [27, 28]. The completeness of the *Australovenator* type specimen has been fundamental to our current understanding of megaraptorid anatomy and phylogenetic hypotheses, and provided robust comparative data that have permitted the assignment of numerous isolated theropod specimens from the mid-Cretaceous of New South Wales [29, 30] and Victoria [1, 9] to Megaraptora (or its subclade Megaraptoridae), and validated an earlier report of a *Megaraptor*-like theropod from Victoria, based on an ulna [31]. The spatiotemporal range of Megaraptora (and Megaraptoridae) is becoming ever better understood as a result of numerous discoveries made within the last two decades. The South American record is the most extensive, diverse and abundant, with six taxa named to date: *Aoniraptor libertatem* [32], *Orkoraptor burkei* [33]; *Megaraptor namunhuaiquii* [34-36]; *Murusraptor barrosaensis* [37-39]; *Aerosteon riocoloradensis* [40]; and *Tratayenia rosalesi* [41]. Numerous fragmentary specimens have also been reported from South America (see [1] supplementary table 7), which include the oldest (Albian [42]) and the youngest (Santonian [41, 43, 44]) occurrences of the clade—unless *Orkoraptor* is truly Coniacian–Campanian in age [45], *contra* previous assessments [10, 46, 47] and more in line with its original reported age [33].

The Asian record of megaraptorans is steadily improving, with occurrences in Japan (*Fukuiraptor kitadaniensis* [48, 49]), China (*Chilantaisaurus tashukouensis* [50, 51] and Thailand (*Phuwiangvenator yaemniyomi* and possibly *Vayuraptor nongbualamphuensis* [52]) (Samathi, Chanthasit & Sander 2019). No megaraptorans are known from Antarctica, Europe or Africa (unless Bahariasauridae is a subclade of Megaraptora [32]) and only one taxon (*Siats meekerorum* [53]) is known from North America. These discoveries — and those made in Australia — have provided a much-improved understanding of megaraptoran skeletal anatomy, thereby facilitating the identification of isolated megaraptorid material. Herein, we describe the fragmentary remains of only the second megaraptorid specimen (excluding shed teeth) from the lower Upper Cretaceous Winton Formation near Winton, central Queensland, Australia. Morphological comparisons aided by three dimensional software imaging were conducted in order to constrain its phylogenetic position, the implications of which are discussed herein.

3. Institutional Abbreviations

Australian Age of Dinosaurs Museum of Natural History, Winton, Queensland, Australia (AAOD);
Australian Age of Dinosaurs Fossil (AODF); Australian Age of Dinosaurs Locality (AODL);
Museums Victoria (formerly National Museum of Victoria), Melbourne, Victoria, Australia (NMV).

4. Geological Setting

1
2
3
4 The Winton Formation is the uppermost unit of the Eromanga Basin, a large continental basin that
5 covers much of western Queensland [54]. The Winton Formation is transitional from the underlying
6 marginal marine Mackunda Formation, with thin lenses of coastal and estuarine deposits persisting in
7 the lower part of the formation [54] and dominated by sand- and mud-dominated facies
8 representative of fluvial conditions in the upper (Cenomanian–lower Turonian) part of the formation
9 [55,56]. The ‘Marilyn’ Site (AODL 261; nicknamed for its proximity to Mount Munro) was
10 discovered and excavated on Elderslie Station in 2018, roughly 500 m west from the *Australovenator*
11 type locality [21]. Rocks in this area are interpreted to come from the uppermost part of the Winton
12 Formation close to the Cenomanian-Turonian boundary [56]. Surface exposures of the local geology
13 are lacking in this area: The majority of specimens collected from this locality were exposed at or
14 close to the surface within the montmorillonite-rich vertisol layer (colloquially termed “black soil”)
15 that blankets the Winton Formation across much of the Winton Shire. Each bone fragment exposed
16 on the surface was flagged prior to collection so that the areal extent of the specimens could be
17 determined: The main concentration of bone occupied an area no more than 15 square metres.
18 Vertebrate remains in this area are naturally exhumed from deeper (i.e. bedrock) layers by convective
19 processes caused by the wetting/drying and the resulting swelling/contraction of the clay-rich soils.
20 Deeper excavations at AODL 261 failed to recover additional remains; the layer presumed to be the
21 source of the dinosaur remains was a ~5–10 cm thick layer of very fine sandy-clay with sporadic
22 reworked plant fossils that were not formally identified. Below the plant-bearing layer was a barren,
23 bluish-grey clay (> 1m thick) entirely devoid of fossils (figure 1). Such clays, including those thought
24 to have been the source of the current specimens, have been interpreted to represent low energy
25 fluvial deposits [4, 21, 23-25, 57-60].
26
27
28
29
30
31
32
33
34
35
36
37
38
39
40
41
42
43
44
45
46
47
48
49
50
51
52
53
54
55
56
57
58
59
60

5. Materials and Methods

The *Australovenator wintonensis* holotype specimens were computed tomography (CT) scanned at Queensland X-ray (Mackay Mater Hospital, Mackay, Queensland) using a Philips Brilliance CT 64-slice machine capable of producing 0.9 mm slices. Mimics version 10.01 (Materialise HQ, Leuven, Belgium) was used to create the 3D surface meshes of the specimens. The meshes were exported as Binary *.stl files into Rhinoceros 5.0 (64-bit; Robert McNeal & Associates, Seattle, WA, USA), which was used to convert the files from *.stl to *.obj file format so they could be imported into Zbrush 4R7 P3 (Pixologic). The fragmentary megaraptorid specimens described herein were scanned using an Artec Space Spider 3D surface scanner.

The resulting 3D scans were exported as *.obj files so that they could be imported into Zbrush 4R7 P3 (Pixologic). Zbrush was used to digitally align and scale these specimens with the corresponding elements in *Australovenator* to confirm initial visual identification.

6. Systematic Palaeontology

Theropoda Marsh, 1881 [61]

Tetanurae Gauthier, 1986 [62]

Megaraptora Benson, Carrano & Brusatte, 2010 [27]

Megaraptora gen. et sp. indet.

6.1 Material

Two incomplete caudal centra (AODF967-968) (figures 2–3), proximal end of metatarsal II (AODF977) (figure 4), distal end of metatarsal II (AODF 978) (figure 5) distal end of metatarsal IV (AODF979) (figure 6), distal end of left pedal phalanx II-1 (AODF972) (figure 7), and numerous unidentified fragments.

6.2 Locality

The ‘Marilyn’ Site (AODL 261), Elderslie Station, ~60 km NW of Winton, Queensland, Australia.

6.3 Horizon and Age

Uppermost Winton Formation, Rolling Downs Group, Eromanga Basin. Cenomanian–lowermost Turonian [55, 56].

7. Results

Specimen descriptions

Vertebrae (AODF 967–968)

The likely positions of AODF 967 and AOD 968 within the vertebral series were estimated by comparisons with other megaraptorids [33, 36, 37, 40, 41, 63] and augmented by *Neovenator salerii* [64], which is considered by some authors as the sister taxon (or one of the successive outgroups) to Megaraptora [27]. Although incomplete, AODF 968 would likely have been longer than it is wide or tall (based in part on the presumed mid-centrum position of the pleurocoel; see below) with a nearly flat (anterior) endplate and no indication of parapophyses. This combination of features is typical of caudal centra but unlike the anteroposteriorly short dorsal vertebrae and opisthocoleous cervical vertebrae of *Neovenator* and megaraptorans [27, 36, 37, 40]. The absence of chevron facets in AODF 968 further identifies it as the anterior part of the centrum. Overall, AODF 968 resembles the caudal vertebrae of the mid caudal region of in *Aerosteon riocoloradensis* (see figure 9B in Sereno [40]) and *Neovenator* (see Plates 18 in Brusatte [63]). AODF 967 is less complete than AOF 968 and lacks the ventral edge of the centrum. Its proportions are therefore equivocal although its larger overall size suggests a more anterior position in the column than AODF 968 (Table 1). The shallowly concave endplate and absence of parapophyses or sacral rib attachment scars eliminate a position in the cervical, anteriormost dorsal or sacral series.

Its relatively small size in comparison to the metapodials (Table 1) suggest that it does not pertain to one of the dorsal vertebrae, which are typically much larger and have a stronger hour-glass shape than the caudal vertebrae (e.g. [36, 41, 63]). A caudal position more anterior than AODF 968 is therefore tenable. For descriptive purposes, AODF 967 is considered the anterior part of the centrum, although we concede that these attributions (i.e. the anterior part of a caudal centrum) are equivocal. AODF 967 constitutes the anterior portion of a vertebral centrum and lacking the ventral margin (figure 2). In posterior and ventral views, the broken surfaces reveal the camerate and camellate internal structures (figure 2A,B,K,L). The anterior articular surface (figure 2C,D) is shallowly concave and, when complete, would have been subcircular. Dorsally, the left and right neurocentral sutures are open (unfused), and their long axes extend anteromedially–posterolaterally (figure 2I,J). The mediolaterally-concave neural canal is widest anteriorly, becoming narrower more posteriorly. Posterior to the articular endplate, the centrum is mediolaterally constricted, which, when complete, would depict an hour-glass shape in ventral view. The right lateral wall (which is more complete than the left side; figure 2E,F) preserves a small fossa approximately mid-height on the centrum, which appears to represent the posterior margin of a plurocoel. The ventral, right lateral and anterior surfaces are incomplete and poorly preserved, obscuring further morphological details of the centrum. AODF 968 comprises the anterior half of a caudal centrum (figure 3). The broken posterior surface reveals camerate and camellate internal structures (figure 3A,B). The anterior articular surface is elliptical (dorsoventrally taller than wide) and shallowly concave (figure 3C,D). The centrum is mediolaterally constricted posterior to the articular endplate whereas the ventral edge (in lateral aspect) is nearly perpendicular to the endplate suggesting that the centrum was notably dorsoventrally constricted. In right lateral view there is a semicircular fossa situated close to the broken posterior edge at roughly two-thirds the height of the centrum and perforated by a pleurocoelous foramen (figure 3E,F). Directly posteromedial to this depression is a camerate internal structure resembling a pleurocoel. In dorsal view (figure 3I,J), the neural canal is mediolaterally concave and posteriorly tapering, bounded on either side by an anterolaterally–posteromedially oriented neurocentral suture.

1 The open sutural surfaces preserve numerous mediolaterally oriented grooves and ridges that would
2 have reinforced the union with the pedicels of the corresponding neural arch. In ventral view (figure
3 3K,L), the centrum is transversely convex with no indication of a ventral groove or keel.
4
5

7 **Proximal end of left metatarsal II (AODF 977)**

8
9
10 The proximal end of a partial left metatarsal II is preserved. The proximal articular surface is
11 somewhat pear shaped (narrowest posteriorly) and nearly flat (figure 4A,B). The proximal part of the
12 metatarsal tapers immediately distal to the proximal articular surface, forming a shaft that is circular
13 in cross section (figure 4C,D), which is similar to the circular shaft in the metatarsal II of
14 *Australovenator* [24] (see figure 7 in White [24]). In medial (figure 4E,F) and lateral (figure 4G,H)
15 views, the posterior margin is extended posteriorly relative to the preserved shaft. The medial margin
16 is incomplete, exposing trabecular bone. The lateral margin is more complete, laterally convex and
17 distomedially inclined, providing an articular surface for metatarsal III.
18
19
20
21
22
23
24
25
26
27
28

29 **Distal end of right metatarsal IV (AODF 979)**

30
31
32 This specimen is interpreted as the distal end of a right metatarsal IV (AODF 979) based on 
33 comparisons with *Neovenator* (see Plate 43 in Brusatte [63]). In distal view (figure 5A,B), the lateral
34 malleolus is inclined ($\sim 70^\circ$) dorsomedially, whereas the medial malleolus is nearly vertical ($\sim 5^\circ$).
35
36 The borders of the medial and lateral collateral ligament pits are heavily eroded; nevertheless, the pits
37 are distinguishable. The lateral malleolus is larger than the medial one and the two are separated by a
38 sulcus (flexor groove), which extends from the posterior (plantar) surface where it is deepest, to the
39 anterior (dorsal) surface where it is comparatively shallow (figure 5A-D). This groove does not
40 extend onto the short section of the preserved shaft nor is there any indication of an extensor pit
41 proximal to the articular surface. The left metatarsal IV is present in the holotype of *Australovenator*
42 (AODF 604) but its distal end is not preserved (see Figures 9 & 10 in White [24], preventing any
43 comparisons between the two.
44
45
46
47
48
49
50
51
52
53
54
55
56
57
58
59
60

Distal end of right metatarsal II (AODF 978)

Based on comparisons with *Australovenator* (see figure 7 in White [24]) and *Neovenator* (see Plate 42 in Brusatte [63]), AODF 978 is interpreted as the incomplete distal end of a right metatarsal II. The distal articular surface is nearly hemispherical but separated posteriorly (ventrally) into subequal medial and lateral malleoli by a broad, flexor groove. This groove is shallow, but likely misrepresented due to breakage and weathering of both medial and lateral malleolus. In anterior (dorsal) view (figure 6A,F), the distal condyle terminates proximally in a lip that borders a prominent extensor pit. In ventral (plantar) view, the medial malleolus extends further proximally than the lateral one. The medial malleolus is incomplete posteriorly and probably would have been somewhat longer still in life (based on comparisons with *Australovenator* (see figure 7 in White [24])). A shallow collateral ligament pit is present on the medial surface (figure 6C,H), whereas the lateral pit is deep but missing part of the ventral rim (figure 6D,I). Digitally superimposing AODF 978 with the distal end of metatarsal II of *Australovenator*, helps to visualise a number of non-trivial differences (figure 6). The distal end of AODF 978 is more hemispherical in dorsal aspect than the strongly asymmetrical metatarsal II of *Australovenator* (figure 6P,Q). More specifically, the medial malleolus of *Australovenator* is proximally positioned relative to the medial malleolus, mediolaterally compressed and bladelike (figure 6T). In contrast, the medial malleolus of AODF 978 falls along the same transverse plane as the lateral malleolus (in posterior aspect; figure 6G) and, despite being incomplete, is relatively robust. In posterior view, the sulcus separating the malleoli is shallower in AODF 978 than in *Australovenator* (figure 6Q), although this may be exaggerated by breakage/weathering in the former. Additionally, AODF 978 is distinctly larger than *Australovenator* (Table 1). Intriguingly, the distal end of metatarsal II (AODF 978) closely resembles the same element (UNPSJB-Pv944) that was tentatively assigned to *Megaraptor* sp. [65] (figure 7) from the roughly coeval Bajo Barreal Formation (Chubut Group, Golfo do San Jorge Basin) of Chubut Province, Argentina.

1
2
3
4
5
6
7
8
9
10
11
12
13
14
15
16
17
18
19
20
21
22
23
24
25
26
27
28
29
30
31
32
33
34
35
36
37
38
39
40
41
42
43
44
45
46
47
48
49
50
51
52
53
54
55
56
57
58
59
60

In particular, both specimens share a distal articular surface that somewhat hemispherical with medial and lateral malleoli that fall along the same transverse plane (or nearly so in the case of UNPSJB-Pv944) in ventral aspect. In distal view, the flexor groove separating the medial and lateral malleoli is relatively shallow (although possibly an artefact, accentuated in AODF 978 by breakage) compared to *Australovenator*. The weathering suffered by AODF 978 precludes any useful comparisons of the medial or lateral surfaces. Unfortunately, a transparent overlay could not be replicated for the UNPSJB-PV 944 specimen as a 3D surface mesh has not yet been developed for the specimen.

Distal end of left pedal phalanx II-1 (AODF 972)

The sole pedal phalanx (AODF 972) recovered from AODL 261 is interpreted as left II-1, based on comparisons with *Australovenator* [24]; however, due to the specimen's incompleteness this identification is tentative. The specimen consists of the distal articular end and a short section of the shaft, which is subcircular in cross section and hollow. The distal articular surface is ginglymous, dorsoventrally and, to a lesser extent, mediolaterally expanded relative to the shaft (figure 8E,J). Although broken, the medial condyle is dorsoventrally shorter than the lateral one but roughly equal in mediolateral width (Table 1). The collateral ligament pits, while present, are infilled with ironstone (figure 8C,D,H,I). This element does not differ notably from that of *Australovenator* (figure 8P–T). Minor areas of morphological discrepancy can be attributed to breakage and/or the adherent ironstone matrix (figure 8A–E).

8. Discussion

Fragmentary theropod remains recovered from the 'Marilyn' Site (AODL 261) constitute only the second theropod specimen (excluding shed teeth) from the Winton Formation. The close proximity and size congruence of the specimens recovered from AODL 261 suggests that they pertain to a single individual.

Unfortunately, our failure to locate further theropod remains in the suspected source layer precludes identification of the taphonomic processes to which these bones were subjected prior to their exposure: the effects of all such processes have been overprinted by much more recent weathering. Identifiable elements are limited in number—two fragmentary vertebrae, three partial metatarsals and the distal end of a pedal phalanx—and all were significantly weathered. The poor preservation and lack of diagnostic features amongst the described specimens prevented a meaningful phylogenetic analysis from being undertaken. Nevertheless, some characters typical of megaraptorids — including camerate and camellate vertebral centra [30, 36] and the presence of pleurocoels [39] — are both evident in AODF 967 and AODF 968. The identification of the AODL 261 material as megaraptorid lies principally on the presence of pleurocoels on the two incomplete caudal centra (although we concede that the caudal position of one of these [AOD 968] is equivocal). Pleurocoels are uncommon in the caudal vertebrae of theropods [33]. Although few megaraptorid caudal vertebrae are known, pleurocoels are present in *Aerosteon* [40], *Megaraptor* [36], and *Orkoraptor* [33] but absent in *Aoniraptor* [32] and possibly the Brazilian megaraptorid SMNS 58023 [42]. Caudal pleurocoels are absent in the immediate outgroups to Megaraptoridae (e.g., *Fukuiraptor* [49], unknown in *Chilantaisaurus*) as well as *Neovenator* [63] but are present in the megalosaurid *Torvosaurus* [Britt 1991] [66], the carcharodontosaurid *Carcharodontosaurus* [67] and oviraptorosaurs, none of which have been unambiguously identified from Australia [3][9]. The distal end of metatarsal II (AODF 978) also bears some resemblance to that of a specimen assigned to *Megaraptor* sp. (UNPSJB-PV 944 [65]) and to a lesser extent *Australovenator* [24]. It alone is not diagnostic enough to identify as a megaraptorid; however, alongside the pleurocoelus caudal vertebrae, its dimensions suggest the individual to which they pertained was slightly larger than the *Australovenator wintonensis* type individual (AODF 604), and that it was possibly similar in size to the largest megaraptorids known from Victoria (NMV P186153) [1, 68] and New South Wales [29]. Our preliminary results indicate that these remains belong to a second megaraptorid taxon from the Winton Formation based primarily on the distal end of metatarsal II; however, more complete and better-preserved material is required to establish this claim.

1
2
3
4
5
6
7
8
9
10
11
12
13
14
15
16
17
18
19
20
21
22
23
24
25
26
27
28
29
30
31
32
33
34
35
36
37
38
39
40
41
42
43
44
45
46
47
48
49
50
51
52
53
54
55
56
57
58
59
60

The occurrence of sympatric megaraptorans within a single formation is potentially predicated by the discovery of *Phuwiangvenator* and *Vayuraptor* from the Sao Khua Formation in Thailand [52].

9. Conclusion

This paper describes the fragmentary remains of only the second non-avian theropod skeleton recovered from the Winton Formation in Central Queensland, Australia. The remains, presumed to have come from a single individual are assigned to Megaraptoridae gen. et sp. indet. based on the presence of camerate and camellate internal structures and the presence of pleurocoels in caudal vertebrae. Given the size of the distal ends of metatarsal II (AODF 972) and IV (AODF 979), this individual would have been larger than the holotype of *Australovenator* (AODF604). Additionally, morphological discrepancies between *Australovenator* and the new specimens highlighted with the use of three dimensional software, may be representative of either ontogenetic/intraspecific variation or indicative of the presence of a second megaraptorid from the Winton Formation. If correct, the latter interpretation adds further support to previous claims that megaraptorids were the dominant large predator in many Australian mid-Cretaceous terrestrial ecosystems.

Acknowledgments

The authors thank the Australian Age of Dinosaurs Museum, Queensland, Australia and Museums Victoria, Melbourne, Australia for access to the specimens. The authors are grateful to all of the Australian Age of Dinosaur participants who attended the 2018 dig season and helped with the excavation. Thanks to Dr Joseph Bevitt and Dr Anton Maksimenko for assistance in data collection at ANSTO. Thanks to Rubén Martínez for giving permission to image Megaraptor sp. (UNPSJB-PV 944) as part of our research. Thanks to the site manager Robert A. Elliott for managing the excavation of dinosaur material and driving the excavation equipment.

Ethical Statement

N/A

Funding Statement

Funding for this work was provided to MAW by the UNE Post-Doctoral Research Fellow Operating Grant provided by the University of New England, Armidale; ANSTO for Imaging and Medical beamline usage award (project ID: AS183/IMBL/13963) to MAW; Australian Research Council Discovery Early Career Researcher Award (project ID: DE170101325) to PRB.

Data Accessibility

The 3D data that support the findings of this study are available from the Dryad Digital Repository: <https://doi.org/10.5061/dryad.4q5j211>

Competing Interests

We have no competing interests.

Authors' Contributions

MAW, PRB designed the project. MAW, PRB, SFP described the specimens. MAW collected synchrotron data. MAW assembled the figures. MAW, PRB, SFP, AHP, SLR, AGC contributed to writing of the paper. MAW, TS, DAE oversaw the collection, preparation and curation of the fossils. MAW, PRB, SFP, AHP, SLR, AGC, TS, DAE approve of the version to be published. MAW, PRB, SFP, AHP, SLR, AGC, TS, DAE agree to be accountable for all aspects of the work in ensuring that questions related to the accuracy or integrity of any part of the work are appropriately investigated and resolved.

References

- Poropat SF, White MA, Vickers-Rich P, Rich TH. in press New megaraptorid (Dinosauria: Theropoda) remains from the Lower Cretaceous Eumeralla Formation of Cape Otway, Victoria, Australia. *Journal of Vertebrate Paleontology*, 2018.
2. Huene Fv. 1932 Die fossile Reptil-Ordnung Saurischia, ihre Entwicklung und Geschichte. *Monographien zur Geologie und Paläontologie (Series 1)* **4**, 1–361.
3. Agnolin FL, Ezcurra MD, Pais DF, Salisbury SW. 2010 A reappraisal of the Cretaceous non-avian dinosaur faunas from Australia and New Zealand: evidence for their Gondwanan affinities. *Journal of Systematic Palaeontology* **8**, 257–300. (doi:10.1080/14772011003594870)
4. White MA, Falkingham PL, Cook AG, Hocknull SA, Elliott DA. 2013 Morphological comparisons of metacarpal I for *Australovenator wintonensis* and *Raptor ornitholestoides*: implications for their taxonomic relationships. *Alcheringa* **37**, 435–441. (doi:10.1080/03115518.2013.770221)
5. Molnar RE, Pledge NS. 1980 A new theropod dinosaur from South Australia. *Alcheringa* **4**, 281–287. (doi:10.1080/03115518008558972)
6. Rauhut OWM. 2005 Post-cranial remains of 'coelurosaurs' (Dinosauria, Theropoda) from the Late Jurassic of Tanzania. *Geological Magazine* **142**, 97–107. (doi:10.1017/S0016756804000330)
7. Barrett PM, Kear BP, Benson RBJ. 2010 Opalized archosaur remains from the Bulldog Shale (Aptian: Lower Cretaceous) of South Australia. *Alcheringa* **34**, 293–301. (doi:10.1080/03115511003664440)
8. Rich TH, Vickers-Rich P. 1994 Neoceratopsians and ornithomimosaur: dinosaurs of Gondwana origin? *National Geographic Research and Exploration* **10**, 129–131.
9. Benson RBJ, Rich TH, Vickers-Rich P, Hall M. 2012 Theropod fauna from southern Australia indicates high polar diversity and climate-driven dinosaur provinciality. *PLOS One* **7**, e37122. (doi:10.1371/journal.pone.0037122)
10. Novas FE, Agnolin FL, Ezcurra MD, Porfiri J, Canale JI. 2013 Evolution of the carnivorous dinosaurs during the Cretaceous: the evidence from Patagonia. *Cretaceous Research* **45**, 174–215. (doi:10.1016/j.cretres.2013.04.001)
11. Delcourt R, Grillo ON. 2018 Tyrannosauroids from the Southern Hemisphere: implications for biogeography, evolution, and taxonomy. *Palaeogeography, Palaeoclimatology, Palaeoecology* **511**, 379–387. (doi:10.1016/j.palaeo.2018.09.003)
12. Poropat SF, Martin SK, Tosolini A-MP, Wagstaff BE, Bean LB, Kear BP, Vickers-Rich P, Rich TH. 2018 Early Cretaceous polar biotas of Victoria, southeastern Australia—an overview of research to date. *Alcheringa* **42**, 157–229. (doi:10.1080/03115518.2018.1453085)
13. Long JA, Molnar RE. 1998 A new Jurassic theropod dinosaur from Western Australia. *Records of the Western Australian Museum* **19**, 121–129.
14. Rauhut OWM. 2007 A fragmentary theropod skull from the Middle Jurassic of Patagonia. *Ameghiniana* **44**, 479–483.
15. Carrano MT, Sampson SD. 2008 The phylogeny of Ceratosauria (Dinosauria: Theropoda). *Journal of Systematic Palaeontology* **6**, 183–236. (doi:10.1017/S1477201907002246)
16. Ezcurra MD, Agnolin FL, Novas FE. 2010 An abelisauroid dinosaur with a non-atrophied manus from the Late Cretaceous Pari Aike Formation of southern Patagonia. *Zootaxa* **2450**, 1–25.
17. Ezcurra MD, Agnolin FL. 2012 An abelisauroid dinosaur from the Middle Jurassic of Laurasia and its

- implications on theropod palaeobiogeography and evolution. *Proceedings of the Geologists' Association* **123**, 500–507. (doi:10.1016/j.pgeola.2011.12.003)
18. Langer MC, Ezcurra MD, Bittencourt JS, Novas FE. 2010 The origin and early evolution of dinosaurs. *Biological Reviews* **85**, 55–110.
19. Fitzgerald EMG, Carrano MT, Holland T, Wagstaff BE, Pickering D, Rich TH, Vickers-Rich P. 2012 First ceratosaurian dinosaur from Australia. *Naturwissenschaften* **99**, 397–405. (doi:10.1007/s00114-012-0915-3)
20. Rauhut OWM. 2012 A reappraisal of a putative record of abelisauroid theropod dinosaur from the Middle Jurassic of England. *Proceedings of the Geologists' Association* **123**, 779–786. (doi:10.1016/j.pgeola.2012.05.008)
21. Hocknull SA, White MA, Tischler TR, Cook AG, Calleja ND, Sloan T, Elliott DA. 2009 New mid-Cretaceous (latest Albian) dinosaurs from Winton, Queensland, Australia. *PLOS One* **4**, e6190. (doi:10.1371/journal.pone.0006190)
22. White MA, Bell PR, Cook AG, Barnes DG, Tischler TR, Bassam BJ, Elliott DA. 2015 Forearm range of motion in *Australovenator wintonensis* (Theropoda, Megaraptoridae). *PLOS One* **10**, e0137709. (doi:10.1371/journal.pone.0137709)
23. White MA, Bell PR, Cook AG, Poropat SF, Elliott DA. 2015 The dentary of *Australovenator wintonensis* (Theropoda, Megaraptoridae); implications for megaraptorid dentition. *PeerJ* **3**, e1512. (doi:10.7717/peerj.1512)
24. White MA, Benson RBJ, Tischler TR, Hocknull SA, Cook AG, Barnes DG, Poropat SF, Wooldridge SJ, Sloan T, Sinapius GHK, et al. 2013 New *Australovenator* hind limb elements pertaining to the holotype reveal the most complete neovenatorid leg. *PLOS One* **8**, e68649. (doi:10.1371/journal.pone.0068649)
25. White MA, Cook AG, Hocknull SA, Sloan T, Sinapius GHK, Elliott DA. 2012 New forearm elements discovered of holotype specimen *Australovenator wintonensis* from Winton, Queensland, Australia. *PLOS One* **7**, e39364. (doi:10.1371/journal.pone.0039364)
26. White MA, Cook AG, Klinkhamer AJ, Elliott DA. 2016 The pes of *Australovenator wintonensis* (Theropoda: Megaraptoridae): analysis of the pedal range of motion and biological restoration. *PeerJ* **4**, e2312. (doi:10.7717/peerj.2312)
27. Benson RBJ, Carrano MT, Brusatte SL. 2010 A new clade of archaic large-bodied predatory dinosaurs (Theropoda: Allosauroidae) that survived to the latest Mesozoic. *Naturwissenschaften* **97**, 71–78. (doi:10.1007/s00114-009-0614-x)
28. Novas FE, Aranciaga Rolando AM, Agnolín FL. 2016 Phylogenetic relationships of the Cretaceous Gondwanan theropods *Megaraptor* and *Australovenator*: the evidence afforded by their manual anatomy. *Memoirs of Museum Victoria* **74**, 49–61.
29. Bell PR, Cau A, Fanti F, Smith E. 2016 A large-clawed theropod (Dinosauria: Tetanurae) from the Lower Cretaceous of Australia and the Gondwanan origin of megaraptorid theropods. *Gondwana Research* **36**, 473–487. (doi:10.1016/j.gr.2015.08.004)
30. Brougham T, Smith ET, Bell PR. 2019 New theropod (Tetanurae: Avetheropoda) material from the 'mid'-Cretaceous Grimman Greek[sic: Creek] Formation at Lightning Ridge, New South Wales, Australia. *Royal Society Open Science* **6**, 180826. (doi:10.1098/rsos.180826)
31. Smith ND, Makovicky PJ, Agnolín FL, Ezcurra MD, Pais DF, Salisbury SW. 2008 A *Megaraptor*-like theropod (Dinosauria: Tetanurae) in Australia: support for faunal exchange across eastern and western Gondwana in the Mid-Cretaceous. *Proceedings of the Royal Society B* **275**, 2085–2093. (doi:10.1098/rspb.2008.0504)
32. Motta MJ, Aranciaga Rolando AM, Rozadilla S, Agnolín FE, Chimento NR, Brissón Egli F, Novas FE. 2016 New theropod fauna from the Upper Cretaceous (Huincul Formation) of northwestern Patagonia, Argentina. *New Mexico Museum of Natural History and Science Bulletin* **71**, 231–253.
33. Novas FE, Ezcurra MD, Lecuona A. 2008 *Orkoraptor burkei* nov. gen. et sp., a large theropod from the Maastrichtian Pari Aike Formation, southern Patagonia, Argentina. *Cretaceous Research* **29**, 468–480. (doi:10.1016/j.cretres.2008.01.001)
34. Novas FE. 1998 *Megaraptor namunhuaiqui*, gen. et sp. nov., a large-clawed, Late Cretaceous theropod from Patagonia. *Journal of Vertebrate Paleontology* **18**, 4–9. (doi:10.1080/02724634.1998.10011030)
35. Calvo JO, Porfiri JD, Veralli C, Novas F, Poblete F. 2004

- Phylogenetic status of *Megaraptor namunhuaiquii* Novas based on a new specimen from Neuquén, Patagonia, Argentina. *Ameghiniana* **41**, 565–575.
36. Porfiri JD, Novas FE, Calvo JO, Agnolin FL, Ezcurra MD, Cerda IA. 2014 Juvenile specimen of *Megaraptor* (Dinosauria, Theropoda) sheds light about tyrannosauroid radiation. *Cretaceous Research* **51**, 35–55. (doi:10.1016/j.cretres.2014.04.007)
37. Coria RA, Currie PJ. 2016 A new megaraptoran dinosaur (Dinosauria, Theropoda, Megaraptoridae) from the Late Cretaceous of Patagonia. *PLOS One* **11**, e0157973. (doi:10.1371/journal.pone.0157973)
38. Paulina-Carabajal A, Currie PJ. 2017 The braincase of the theropod dinosaur *Murusraptor*: osteology, neuroanatomy and comments on the paleobiological implications of certain endocranial features. *Ameghiniana* **54**, 617–640. (doi:10.5710/AMGH.25.03.2017.3062)
39. Aranciaga Rolando AM, Novas FE, Agnolin FL. 2019 A reanalysis of *Murusraptor barrosaensis* Coria & Currie (2016) affords new evidence about the phylogenetical relationships of Megaraptora. *Cretaceous Research* **99**, 104–127. (doi:10.1016/j.cretres.2019.02.021)
40. Sereno PC, Martinez RN, Wilson JA, Varricchio DJ, Alcober OA, Larsson HCE. 2008 Evidence for avian intrathoracic air sacs in a new predatory dinosaur from Argentina. *PLOS One* **3**, e3303. (doi:10.1371/journal.pone.0003303)
41. Porfiri JD, Juárez Valieri RD, Santos DDD, Lamanna MC. 2018 A new megaraptoran theropod dinosaur from the Upper Cretaceous Bajo de la Carpa Formation of northwestern Patagonia. *Cretaceous Research* **89**, 302–319. (doi:10.1016/j.cretres.2018.03.014)
42. Aranciaga Rolando AM, Brissón Egli F, Sales MAF, Martinelli AG, Canale JI, Ezcurra MD. 2018 A supposed Gondwanan oviraptorosaur from the Albian of Brazil represents the oldest South American megaraptoran. *Cretaceous Research* **84**, 107–119. (doi:10.1016/j.cretres.2017.10.019)
43. Sales MAF, Martinelli AG, Francischini H, Rubert RR, Marconato LP, Soares MB, Schultz CL. 2017 New dinosaur remains and the tetrapod fauna from the Upper Cretaceous of Mato Grosso State, central Brazil. *Historical Biology*, (doi:10.1080/08912963.2017.1315414)
44. Gianechini FA, Lio G, Apesteguía S. 2011 Isolated archosaurian teeth from “La Bonita” locality (Late Cretaceous, Santonian–Campanian), Río Negro Province, Argentina. *Historia Natural* **1**, 5–16.
45. Sickmann ZT, Schwartz TM, Graham SA. 2018 Refining stratigraphy and tectonic history using detrital zircon maximum depositional age: an example from the Cerro Fortaleza Formation, Austral Basin, southern Patagonia. *Basin Research* **30**, 708–729. (doi:10.1111/bre.12272)
46. Ezcurra MD, Novas FE. 2016 Theropod dinosaurs from Argentina. In *Historia evolutiva y paleobiogeográfica de los vertebrados de América del Sur*. (ed. eds. F. L. Agnolin, G. L. Lio, F. Brissón Egli, N. R. Chimento, F. E. Novas), pp. 139–156. Buenos Aires: Museo Argentino de Ciencias Naturales “Bernardino Rivadavia”.
47. O’Gorman JP, Varela AN. 2010 The oldest lower Upper Cretaceous plesiosaurs (Reptilia, Sauropterygia) from southern Patagonia, Argentina. *Ameghiniana* **47**, 447–459. (doi:10.5710/AMGH.v47i4.3)
48. Azuma Y, Currie PJ. 2000 A new carnosaur (Dinosauria: Theropoda) from the Lower Cretaceous of Japan. *Canadian Journal of Earth Sciences* **37**, 1735–1753.
49. Currie PJ, Azuma Y. 2006 New specimens, including a growth series, of *Fukuiraptor* (Dinosauria, Theropoda) from the Lower Cretaceous Kitadani Quarry of Japan. *Journal of the Paleontological Society of Korea* **22**, 173–193.
50. Hu S-y. 1964 Carnosaurian remains from Alashan, Inner Mongolia. *Vertebrata Palasiatica* **8**, 42–63.
51. Benson RBJ, Xu X. 2008 The anatomy and systematic position of the theropod dinosaur *Chilantaisaurus tashuikouensis* Hu, 1964 from the Early Cretaceous of Alanshan, People’s Republic of China. *Geological Magazine* **145**, 778–789. (doi:10.1017/S0016756808005475)
52. Samathi A, Chanthasit P, Sander PM. 2019 Two new basal coelurosaurian theropod dinosaurs from the Lower Cretaceous Sao Khua Formation of Thailand. *Acta Palaeontologica Polonica* **64**, 239–260. (doi:10.4202/app.00540.2018)
53. Zanno LE, Makovicky PJ. 2013 Neovenatorid theropods are apex predators in the Late Cretaceous of

- North America. *Nature Communications* **4**, 1–9. (doi:10.1038/ncomms3827)
54. Cook AG, Bryan SE, Draper JJ. 2013 Post-orogenic Mesozoic basins and magmatism. In *Geology of Queensland*. (ed. ^eds. P. A. Jell), pp. 515–575. Brisbane: Geological Survey of Queensland.
55. Bryan SE, Cook AG, Allen CM, Siegel C, Purdy DJ, Greentree JS, Uysal IT. 2012 Early–mid Cretaceous tectonic evolution of eastern Gondwana: from silicic LIP magmatism to continental rupture. *Episodes* **35**, 142–152.
56. Tucker RT, Roberts EM, Hu Y, Kemp AIS, Salisbury SW. 2013 Detrital zircon age constraints for the Winton Formation, Queensland: contextualizing Australia’s Late Cretaceous dinosaur faunas. *Gondwana Research* **24**, 767–779. (doi:10.1016/j.gr.2012.12.009)
57. Rey PF. 2013 Opalisation of the Great Artesian Basin (central Australia): an Australian story with a Martian twist. *Australian Journal of Earth Sciences* **60**, 291–314. (doi:10.1080/08120099.2013.784219)
58. Poropat SF, Mannion PD, Upchurch P, Hocknull SA, Kear BP, Elliott DA. 2015 Reassessment of the non-titanosaurian somphospondylan *Wintonotitan watsi* (Dinosauria: Sauropoda: Titanosauriformes) from the mid-Cretaceous Winton Formation, Queensland, Australia. *Papers in Palaeontology* **1**, 59–106. (doi:10.1002/spp2.1004)
59. Poropat SF, Upchurch P, Mannion PD, Hocknull SA, Kear BP, Sloan T, Sinapius GHK, Elliott DA. 2015 Revision of the sauropod dinosaur *Diamantinasaurus matildae* Hocknull et al. 2009 from the middle Cretaceous of Australia: implications for Gondwanan titanosauriform dispersal. *Gondwana Research* **27**, 995–1033. (doi:10.1016/j.gr.2014.03.014)
60. White MA, Cook AG, Rumbold SJ. 2017 A methodology of theropod print replication utilising the pedal reconstruction of *Australovenator* and a simulated paleo-sediment. *PeerJ* **5**:e3427 (doi.org/10.7717/peerj.3427)
61. Marsh OC. 1881 Principal characters of American Jurassic dinosaurs. Part V. *American Journal of Science* **21 (series 3)**, 417–423. (doi:10.2475/ajs.s3-21.125.417)
62. Gauthier J. 1986 Saurischian monophyly and the origin of birds. In *The Origin of Birds and the Evolution of Flight: Memoirs of the California Academy of Sciences*, **8**. (ed. ^eds. K. Padian), pp. 1–55
63. Brusatte SL, Benson RBJ & Hutt S. 2008. The osteology of *Neovenator salerii* (Dinosauria: Theropoda) from the Wealden Group (Barremian) of the Isle of Wight. Other. Palaeontographical Society, *Palaeontographical Society Monographs* **162** (631).
64. Hutt S, Martill DM, & Barker MJ. 1996. The first European allosauroid dinosaur (Lower Cretaceous, Wealden Group, England). *Neues Jahrbuch für Geologie und Paläontologie Monatshefte* **10**, 635–644.
65. Lamanna MC. 2004. Late Cretaceous dinosaurs and crocodyliforms from Egypt and Argentina: University of Pennsylvania.
66. Britt B. 1991. Theropods of Dry Mesa Quarry (Morrison Formation, Late Jurassic), Colorado, with emphasis on the osteology of *Torvosaurus tanneri*. *Brigham Young University Geology Studies* **37**, 1–72.
67. Novas FE, Valais S, Vickers-Rich P, Rich T. 2005. A large Cretaceous theropod from Patagonia, Argentina, and the evolution of carcharodontosaurids. *Naturwissenschaften* **92**, 226–230.
68. White MA, Poropat SF, Campione NE., *in preparation* A new approach for visualising pairwise morphological differences in three dimensions and its application to isolated palaeontological specimens. *PeerJ*,

Table 1. Selected postcranial measurements of megaraptorid remains from AODL261 compared to *Australovenator*.

Specimens with broken or worn edges (representing incomplete measurements) are marked with an asterisk (*).

Specimens	AODF967 (Fig 2)	AODF968 (Fig 3)	AODF977 (Fig 4)	AODF979 (Fig 5)	AODF978 (Fig 6) Metatarsal II	Austrlovenat or holotype AODF604 (Fig 6) Metatarsal II	AODF972 (Fig 7) MT II-1	Austrlovenat or holotype AODF604 (Fig 7) MT II-1
Centrum width at narrowest point	30*	23						
Centrum height (to neurocentral suture)	45*	41						
Centrum width (anterior end)	46	36						
Proximal width			45*					
Proximal height			67*					
Distal malleolus height (medial, lateral)				41*, 42*	44*, 45*	42, 40	30*, 35*	37, 33
Distal width (measured ventrally)				57*	50*	46	-	43

Figures

For final submissions, figures should be uploaded as separate files.

Figure and table captions

Figure 1. Locality and geological setting of AODL 261 (the ‘Marilyn’ Site). (A) Location of Elderslie Station (star) within the context of the Eromanga Basin (green), Central West Queensland, Australia. (B) Aerial photograph of AODL 261. (C) Schematic interpretation of the sub-surface stratigraphy of AODL 261. Here, fossils are naturally brought to the surface from deeper fossiliferous horizons by the expansion-contraction of the clay-rich soils.

Figure 2. Megaraptorid caudal centrum (AODF 967) in (A,B) anterior, (C,D) posterior (E,F) right lateral, (G,H) left lateral (I,J) dorsal, (K,L) ventral views. Abbreviations: car, camerate internat structure; cam, camellate internal structure; nc, neural canal; p, pleurocoel.

Figure 3. Megaraptorid caudal vertebra (AODF 968) in (A,B) posterior, (C,D) anterior, (E,F) right lateral, (G,H) left lateral, (I, J) dorsal, and (K,L) ventral views. Abbreviations: car, camerate internal structure; cam, camellate internal structure; nc, neural canal; p, pleurocoel.

17 **Figure 4.** Megaraptorid proximal left metatarsal II (AODF 977) in (A,B) proximal, (C,D) distal, (E,F) medial (G,H)
18 lateral views.
19

52 **Figure 5.** Megaraptorid distal right metatarsal IV (AODF 979) In (A,B) distal, (C,D) anterior, (E,F) posterior, (G,H)
53 lateral, and (I,J) medial views. Missing parts are reconstructed with a dashed line.
54
55
56
57
58
59
60

Figure 6. Megaraptorid distal right metatarsal II (AODF 978) compared with the right metatarsal II of *Australovenator wintonensis* (AODF 604). Photographs (A–E) and digital renders (F–J) of megaraptorid right metatarsal II (AODF 978) in (A,F) anterior, (B,G) posterior, (C,H) medial, (D,I) lateral and (E,J) distal views. Digital renders (K–O) of *Australovenator wintonensis* right metatarsal II (AODF 604) in (K) anterior, (L) posterior, (M) medial, (N) lateral and (O) distal views. Digital comparison (P–T) of right second metatarsals of AODF 978 (solid tan) with AODF 604 (*Australovenator*; transparent grey) corrected for scale and orientation in (P) anterior, (Q) posterior, (R) medial, (S) lateral and (T) distal views.

Figure 7. Megaraptorid distal right metatarsal II (AODF 978) compared with distal right metatarsal II of *Megaraptor* sp. (UNPSJB-PV 944). Photographs (A–E) of megaraptorid right metatarsal II (AODF 978) in (A) anterior, (B) posterior, (C) medial, (D) lateral and (E) distal views. Photographs (F–J) of *Megaraptor* sp. right metatarsal II (UNPSJB-PV 944) in (F) anterior, (G) posterior, (H) medial, (I) lateral and (J) distal views.

Figure 8. Megaraptorid distal left pedal phalanx II-1 (AODF 972) compared with left pedal phalanx II-1 of *Australovenator wintonensis* (AODF 604). Photographs (A–E) and digital renders (F–J) of megaraptorid left pedal phalanx II-1 (AODF 972) in (A,F) anterior, (B,G) posterior, (C,H) medial, (D,I) lateral and (E,J) distal views. Digital renders (K–O) of *Australovenator wintonensis* left pedal phalanx II-1 (AODF 604) in (K) anterior, (L) posterior, (M) medial, (N) lateral and (O) distal views. Digital comparison (P–T) of left pedal phalanges II-1 of AODF 978 (solid tan) with AODF 604 (*Australovenator*; transparent grey) in (P) anterior, (Q) posterior, (R) medial, (S) lateral and (T) distal views.

Locality and geological setting of AODL 261 (the 'Marilyn' Site). (A) Location of Elderslie Station (star) within the context of the Eromanga Basin (green), Central West Queensland, Australia. (B) Aerial photograph of AODL 261. (C) Schematic interpretation of the sub-surface stratigraphy of AODL 261. Here, fossils are naturally brought to the surface from deeper fossiliferous horizons by the expansion-contraction of the clay-rich soils.

Megaraptorid ?caudal centrum (AODF 967) in (A,B) anterior, (C,D) posterior (E,F) right lateral, (G,H) left lateral (I,J) dorsal, (K,L) ventral views. Abbreviations: car, camerate internat structure; cam, camellate internal structure; nc, neural canal; p, pleurocoel.

34
35
36
37
38
39
40
41
42
43
44
45
46
47
48
49
50
51
52
53
54
55
56
57
58
59
60

Megaraptorid caudal vertebra (AODF 968) in (A,B) posterior, (C,D) anterior, (E,F) right lateral, (G,H) left lateral, (I, J) dorsal, and (K,L) ventral views. Abbreviations: car, camerate internal structure; cam, camellate internal structure; nc, neural canal; p, pleurocoel.

Megaraptorid proximal left metatarsal II (AODF 977) in (A,B) proximal, (C,D) distal, (E,F) medial (G,H) lateral views.

Megaraptorid distal right metatarsal IV (AODF 979) In (A,B) distal, (C,D) anterior, (E,F) posterior, (G,H) lateral, and (I,J) medial views. Missing parts are reconstructed with a dashed line.

Megaraptorid distal right metatarsal II (AODF 978) compared with the right metatarsal II of *Australovenator wintonensis* (AODF 604). Photographs (A–E) and digital renders (F–J) of megaraptorid right metatarsal II (AODF 978) in (A,F) anterior, (B,G) posterior, (C,H) medial, (D,I) lateral and (E,J) distal views. Digital renders (K–O) of *Australovenator wintonensis* right metatarsal II (AODF 604) in (K) anterior, (L) posterior, (M) medial, (N) lateral and (O) distal views. Digital comparison (P–T) of right second metatarsals of AODF 978 (solid tan) with AODF 604 (*Australovenator*; transparent grey) corrected for scale and orientation in (P) anterior, (Q) posterior, (R) medial, (S) lateral and (T) distal views.

Megaraptorid distal right metatarsal II (AODF 978) compared with distal right metatarsal II of Megaraptor sp. (UNPSJB-PV 944). Photographs (A–E) of megaraptorid right metatarsal II (AODF 978) in (A) anterior, (B) posterior, (C) medial, (D) lateral and (E) distal views. Photographs (F–J) of Megaraptor sp. right metatarsal II (UNPSJB-PV 944) in (F) anterior, (G) posterior, (H) medial, (I) lateral and (J) distal views.

Megaraptorid distal left pedal phalanx II-1 (AODF 972) compared with left pedal phalanx II-1 of *Australovenator wintonensis* (AODF 604). Photographs (A–E) and digital renders (F–J) of megaraptorid left pedal phalanx II-1 (AODF 972) in (A,F) anterior, (B,G) posterior, (C,H) medial, (D,I) lateral and (E,J) distal views. Digital renders (K–O) of *Australovenator wintonensis* left pedal phalanx II-1 (AODF 604) in (K) anterior, (L) posterior, (M) medial, (N) lateral and (O) distal views. Digital comparison (P–T) of left pedal phalanges II-1 of AODF 978 (solid tan) with AODF 604 (*Australovenator*; transparent grey) in (P) anterior, (Q) posterior, (R) medial, (S) lateral and (T) distal views.

Appendix B

Dear Lianne Parkhouse,

Please find my response to the reviewers' comments below,

Response to Reviewers:

Reviewer 1: Changes have been made with tracked changes and disagreements have been noted below.

“PAGE 9 – Lines 31 and 32. The bone illustrated on figure 5 of present ms, seems fairly symmetrical in distal view, being similar to mtt III as well as to some non-ungual pedal phalanges. On the contrary, metatarsal IV in theropods is beveled in distal view. Please, have a look at Calvo et al., 2004 paper (your reference number 35), figure 10 A,C, illustrating the mtt IV of *Megaraptor namunhuaiquii*. Also, consider the possibility that this bone corresponds to a pedal phalanx”

I am familiar with the Calvo et al. 2004 publication however the diagrams provided in this manuscript are extremely poor. The specimen we describe and figure is definitely not symmetrical. The eroded sections as depicted in our figure demonstrate its asymmetrical morphology. Therefore, we have disregarded this recommendation.

PAGE 10. Lines 51-54. Are you meaning that the specimen here described resembles more *Megaraptor* sp from Patagonia, rather than to *Australovenator*? Please, clarify. However, and based on the figures here afforded, distal end of mtt II of *Australovenator* matches well with that of *Megaraptor* sp. from Patagonia, both being different from the new *Megaraptoridae* from Winton in the ventral projection of both inner and outer condyles. Then, let me ask whether shape “distinctions” of the later one are due to erosion, rather than true anatomical features.

The first part of this is addressed “...Formation (Chubut Group, Golfo do San Jorge Basin) of Chubut Province, Argentina **rather than *Australovenator*.**”

However the second portion was already stated “The weathering suffered by AODF 978 precludes any useful comparisons of the medial or lateral surfaces”.

Reviewer 2:

I have made the recommended changes to the manuscript.

Regards

Dr Matt A White